# Evaluation of Doppler radar and GTS Data Assimilation for NWP Rainfall Prediction of an Extreme Summer Storm in Northern China: from the Hydrological Perspective

Jia Liu[1], Jiyang Tian[1], Denghua Yan[1], Chuanzhe Li[1], Fuliang Yu[1], Feifei Shen[2]

[1] State Key Laboratory of Simulation and Regulation of Water Cycle in River Basin, China Institute of Water Resources and Hydropower Research, Beijing, 100038, China

[2] Key Laboratory of Meteorological Disaster of Ministry of Education, Nanjing University of Information Science & Technology, Nanjing, 210044, China

*Correspondence to*: Jia Liu (hettyliu@126.com)

**Abstract.** Data assimilation is an effective tool in improving high-resolution rainfall of the numerical weather prediction (NWP) systems which always fails in providing satisfactory rainfall products for hydrological use. The aim of this study is to explore the potential effects of assimilating different sources of observations, i.e. the Doppler weather radar and the Global Telecommunication System (GTS) data, in improving the mesoscale NWP rainfall products. A 24 h summer storm occurring over the Beijing-Tianjin-Hebei region of northern China on 21 July 2012 is selected as a case study. The Weather Research and Forecasting (WRF) model is used to obtain 3 km rainfall forecasts, and the observations are assimilated using the three-dimensional variational (3D-Var) data assimilation method. Eleven data assimilation modes are designed for assimilating different combinations of observations in the two nested domains of the WRF model. Both the rainfall accumulative amount and its distribution in space and time is examined for the forecasting results with/without data assimilation. The results show that data assimilation can effectively help improve the WRF rainfall forecasts, which is of great importance for hydrologic applications through the rainfall-runoff transformation process. Both the radar reflectivity and the GTS data are good choices for assimilation in improving the rainfall products, whereas special attentions should be paid for assimilating radial velocity where unsatisfactory results are always found. The assimilation of the GTS data in the coarser domain have positive effects on the radar data assimilation in the finer domain, which can make the rainfall forecasts more accurate than assimilating the radar data alone. It is also found that the assimilation of more observations cannot guarantee further improvement of the rainfall products, whereas the effective information contained in the assimilated data is of more importance than the data quantity. Potential improvements of data assimilation in improving the NWP rainfall products are discussed and suggestions are further made.

## 1 Introduction

Numerical weather prediction (NWP) systems play an important role in the prediction of meteorological and hydrological processes with the ability of providing relatively reliable products for analyzing and forecasting weather events (Rodwell et al, 2010; De et al, 2011; Boussetta et al, 2013). Rainfall is not only a crucial meteorological variable but is also a hydrological element; therefore, it is always important to obtain accurate rainfall information for hydrological use. Unfortunately, because of the uncertainties and complexities of atmospheric processes, rainfall is among the most difficult variables to be accurately captured using NWP (Berenguer et al, 2012; Shrestha et al, 2013). The Weather Research and Forecasting (WRF) model is a latest generation mesoscale NWP system that has been widely used for rainfall simulation and prediction (Efstathiou et al, 2013; Yang et al, 2015). Although rainfall products can be directly used due to the high accuracy of rain or no rain predictions, the WRF model still cannot ensure accurate rainfall quantities or spatiotemporal distributions at the catchment scale for hydrological prediction (Liu et al, 2012; Li et al, 2013).

Qie et al. (2014) simulated a storm event that featured short-term (12 h) heavy rainfall and frequent lightning activity over northern China using the WRF model. The results showed that the mean absolute error of the 6 h accumulated rainfall was 39.8 mm for an observed rainfall range of 20-50 mm, whereas the mean absolute error reached 53.3 mm for an observed rainfall range of 11-20 mm. Hamill (2014) used an ensemble prediction system to analyze the performance of the WRF model in northern Colorado. Although ensemble prediction could avoid the complications induced by uncertainties, the accumulated rainfall also had a large bias compared with the observations. Kryza et al. (2013) found that the WRF model always underestimated the 24 h accumulated rainfall in Poland, and the rainfall coverage area greatly changed based on different physical parameterizations. Efstathiou et al. (2013) indicated that the WRF model could not capture the maximum rainfall intensity in either time or space at Chalkidiki, Greece. Gascón et al. (2016) used the WRF model to simulate an exceptionally heavy convective rainfall on 3 July 2006 in Calabria, Italy. The simulation results were unsatisfactory, and the total rainfall was significantly underestimated.

Data assimilation has been shown to be an efficient way in improving the quality of the WRF rainfall products (Liu et al, 2013). The WRF model provides a three dimensional-variational data assimilation system, i.e., WRF-3DVar (Barker et al, 2004), that works in tandem with the WRF model in real-time. The system can assimilate various types of conventional and non-conventional data, such as observations from surface weather stations, buoys, ships, pilot balloons, radars and satellites (Routray et al, 2010). Many studies have shown that WRF-3DVar works well with observations from different sources. Routray et al. (2012) assimilated upper-air and surface data from the Global Telecommunication System (GTS) using WRF-3DVar, and the results showed that the location and amount of rainfall was captured better over the west coast of India when compared with the simulation without data assimilation. Wind observations were also assimilated using WRF-3DVar as the main observed data for rainfall simulation in India, and the rainfall intensity and spatial distribution were considerably improved (Mohanty et al, 2012). To improve heavy rainfall forecasts over the Korean Peninsula, global positioning system

(GPS) radio occultation (RO) data were used with WRF-3DVar (Ha et al, 2014). Results indicated that the quantitative accuracy of the rainfall forecast was in better agreement with observations, especially the maximum rainfall amount.

Compared with other observational data, Doppler radar can obtain detailed rainfall information with spatial resolutions of a few kilometers and temporal resolutions of a few minutes (Sugimoto et al, 2009). With high spatial and temporal resolutions, Doppler radar can reveal detailed structural features of mesoscale storms and capture rapidly developing convective weather systems (Pu et al, 2009; Maiello et al, 2014). Moreover, Doppler radar can provide real-time observations that can be assimilated using WRF-3DVar and used for real-time rainfall forecasts (Liu et al, 2013). Some studies have also indicated that significant improvements could be obtained in rainfall simulations using data assimilation (Bauer et al, 2015), because radar data can provide more detailed information on the initial fields and improve the lateral boundary conditions of the WRF model (Sokol and Pešice, 2009).

Doppler radar can provide two types of observations for assimilation, i.e., radar reflectivity and radial velocity. Both have been found to have positive impacts on NWP rainfall simulations and forecasts, especially for heavy rainfall events (Li et al, 2012). Based on the operating principle of Doppler radar, reflectivity contains information on the number of falling drops per unit volume, which depends on the hydrometeor number and size, whereas radial velocity is related to vertical atmospheric motions (Maiello et al, 2014). This means that assimilating reflectivity impacts the thermodynamic and dynamic fields, whereas radial velocity assimilation only influences the dynamic fields (Xiao and Sun, 2007; Abhilash et al, 2012). Li et al. (2012) indicated that assimilating radial velocity every 30 min could improve the accuracy of rainfall (caused by hurricane) intensity prediction. Sun et al. (2012) found that the pattern and location of forecasted rainfall were noticeably improved with radar reflectivity assimilation. Pan et al. (2012) found that the WRF model can capture rainfall distributions in time and space better by assimilating both Doppler radar reflectivity and radial velocity. Abhilash et al. (2012) went further and compared the assimilation effects of radar reflectivity and radial velocity. Results showed that the assimilation of both radar reflectivity and radial velocity significantly improved most meteorological elements, including wind, moisture and rainfall.

However, it has been found that data assimilation is mainly used for synoptic analyses of meteorological fields. The potential of data assimilation has not been fully studied for hydrological purposes, which require a more rigorous evaluation of rainfall quantities and variations (Liu et al, 2015). Rainfall prediction is especially important for real-time flood forecasting in small sized catchments which often have short concentration times and need predicted rainfall to extend the flood forecast lead time (Liu and Han, 2013). Therefore, hydrologists are particularly concerned about the accuracy of the accumulative amount and the spatiotemporal distribution of the predicted rainfall at the catchment scale. The spatiotemporal distribution of the predicted rainfall directly impacts the forecasted discharge and the timing of the flood peak through rainfall-runoff transformation modeling. To what extent can the assimilation of Doppler radar observations help improve rainfall predictions of particular storm events at the catchment scale? Is it always the best choice to simultaneously assimilate radar reflectivity and radial velocity? In general, Doppler radar is easily contaminated by non-weather returns, such as second-trips, sidelobe clutter, gound clutter and low signal-to-noise returns, which can affect the quality of radar data and the assimilation

effect (Zhang et al, 2012). Therefore, it may be of interest to examine whether rainfall predictions can be improved by assimilating radar data with other observations, such as meteorological elements from fixed and mobile stations.

A 24 h storm event that occurred in July of 2012 in the Beijing-Tianjin-Hebei region of northern China was selected for this study. The storm event has received wide attention due to its high intensity and the significant losses caused by the corresponding flooding. The 24 h rainfall that accumulated in the small mountainous catchment of Zijingguan with the drainage area of 1760 km$^2$, was regenerated using the WRF model with different data assimilation scenarios. Observations from an S-band Doppler weather radar that completely covers the Zijingguan catchment were assimilated with the assistance of WRF-3DVar. Traditional meteorological data from the GTS were obtained from the National Center of Atmospheric Research (NCAR). Eleven different data assimilation modes were designed for assimilating different combinations of the three types of data, i.e., radar reflectivity, radial velocity and GTS observations, in the two nested domains of the WRF model. The improvements in the forecasted rainfall from the 11 data assimilation scenarios were evaluated from the perspectives of the accumulative process, the spatiotemporal distribution and the total amount. The performances of different data assimilation scenarios were compared, and the efficient way to assimilate Doppler radar observations for rainfall prediction was further discussed.

## 2 The WRF model and 3D-Var data assimilation

### 2.1 The WRF model setup

The WRF model (version 3.6) is a next-generation meteorological model that includes a variety of physical options and can be used over a wide range of scales, ranging from tens of meters to thousands of kilometers. Detailed descriptions of the model were available in Skamaraock et al. (2008). Two nested domains were centered over the Zijingguan catchment, and two-way nesting was used for communication between the parent and child domains. To obtain high-resolution rainfall products and make the results applicable for hydrological forecasting systems, the horizontal grid spacing of the WRF inner domain (Domain 2) was set to 3 km, and the downscaling ratio was set to 1:3, which was commonly used and always performed well in the Beijing-Tianjin-Hebei region of northern China (Liu et al, 2012; Chambon et al, 2014; Tian et al, 2017b). The nested domain sizes were 1260×1260 and 450×360 km$^2$ for the Zijingguan catchment. The settings of the nested domains have been shown to be effective in previous studies in terms of representing the large scale topography and the main climate characteristics in the study area (Wang et al, 2013; Tian et al, 2017b). The two domains were comprised of 40 vertical pressure levels, with the top level set to 50 hPa (Done et al., 2004; Aligo et al, 2009; Fierro et al., 2013; Qie et al, 2014). The model initial and lateral boundary conditions were obtained from Global Forecast System (GFS) forecast data, which were provided by NCEP with 1°×1° grids and were widely used to forecast historical storm events (Routray et al, 2010; Ha and Lee, 2012). The time step of the WRF model output was set to one hour. Considering the use of downscaling and the high resolution needed for meteorological and hydrological studies, the forecasted rainfall in the inner domain was chosen for analysis.

Cumulus physics, microphysics and planetary boundary layer (PBL) are important physical parameterization options for rainfall simulations (Cassola et al, 2015; Fernández-González et al, 2015). According to our previous investigations on the performances of the most important WRF physical parameterizations affecting the rainfall processes in Northern China (Tian et al, 2017a and 2017c), the most appropriate set of parameterizations for this extreme summer storm, including Kain-Fritsch (KF), WRF single-moment 6 (WSM6) and Mellor-Yamada-Janjic (MYJ), was adopted in this study when configuring the WRF model (Miao et al, 2011; Guo et al, 2014; Di et al, 2015).

## 2.2 3-DVar data assimilation

The 3-DVar data assimilation produces an optimal estimate of the atmospheric state through iterative solution of a prescribed cost function (Ide et al, 1997):

$$J(x) = \frac{1}{2}(x - x^b)^T B^{-1}(x - x^b) + \frac{1}{2}(y - y^0)^T R^{-1}(y - y^0) \tag{1}$$

where $x$ is the atmospheric and surface state vector, $x^b$ is the first guess or background, and $y^0$ is the assimilated observation. $y$ is the observation space derived from the model. **B** and **R** are the background error covariance matrix and the observation error covariance matrix, respectively.

The WRF-3DVar data assimilation system in the WRF model was used for assimilating the GTS and weather radar data in real-time (Barker et al, 2004; Gao et al, 2004). The background error covariance CV3 was used in this study. The greatest advantage of CV3 is its wide applicability (Meng and Zhang, 2008). The reason why CV3 is used also includes the trial to simplify the data assimilation procedure which enables a more extensive application among hydrologists.

## 2.3 The observation operators: radar reflectivity and radial velocity

In the WRF-3DVar system, the total water mixing ratio $q_t$ was used as the moisture control variable instead of the pseudo-relative humidity when assimilating the radar reflectivity data (Dudhia, 1989). The water mixing ratio has a more direct relation with the radar reflectivity, as described by Eq. (2), which has been proven to be effective in WRF-3DVar as an observation operator to calculate the model-derived radar reflectivity $Z$ from the rainwater mixing ratio $q_r$ (Sun and Crook, 1997).

$$Z = 43.1 + 17.5\log(\rho q_r) \tag{2}$$

where $\rho$ is the density of air. By assuming a Marshall-Palmer raindrop size distribution and that the ice phases have no effect on reflectivity, Eq. (2) can be derived.

For the assimilation of the radial velocity, the preconditioned wind control variables were also combined with the rainwater mixing ratio $q_r$. Eq. (3)-(5) show how the model-derived radial velocity $V_r$ was calculated:

$$V_r = u\frac{x - x_i}{r_i} + v\frac{y - y_i}{r_i} + (w - v_t)\frac{z - z_i}{r_i} \tag{3}$$

where $u$, $v$ and $w$ represent the three-dimensional wind field, $x$, $y$, and $z$ represent the location of the observation point, and $x_i$, $y_i$, and $z_i$ represent the location of the radar station. $r_i$ is the distance between the data point and the radar, and $v_t$ is the hydrometer fall speed or terminal velocity.

According to Sun and Crook (1998), $v_t$ can be represented as follows:

$$v_t = 5.40a(\rho q_r)^{0.125} \tag{4}$$

$$a = \left(\frac{p_0}{\bar{p}}\right)^{0.4} \tag{5}$$

where a is the correction factor, $\bar{p}$ is the base-state pressure, and $p_0$ is the pressure at the ground.

## 3 Study area and data

### 3.1 Study area and the storm event

The Zijingguan catchment, which lies in the northern reach of the Daqing river basin, was chosen as the study area (Fig. 1). It is located at 39°13′~ 39°40′ north latitude and 114°28′~115°11′ east longitude and has a drainage area of 1760 km². It is 54 km long from north to south and 61 km wide from east to west. The Zijingguan catchment has a temperate continental monsoon climate. The average annual rainfall is approximately 600 mm, and the majority of rain falls during the flood season from late May to early September. The size and terrain together with the previous history of extreme storms and

floods make the Zijingguan catchment representative of the catchments in the semi-humid and semi-arid area of northern China that require flood warnings. To avoid enormous losses caused by floods, accurate rainfall forecasts are very important.

**[Figure 1]**

A 24 h storm event that occurred over the Beijing-Tianjin-Hebei region on 21 July 2012 was chosen for this study. Because of the high intensity rainfall, wide coverage and significant losses, the storm event has received widespread attention in

China. Many studies have investigated the causes and the properties of the storm (Sang et al, 2013; Zhong et al, 2015). Before the extreme storm event took place, an encounter between a northward-moving subtropical high vortex and an eastward-moving cold vortex in the mid-high troposphere provided a stable atmospheric circulation over the study area, which was conducive to heavy rain formation. Abundant water vapor transported from a low-level jet, strong upward motion caused by Taihang Mountain, together with a long duration of dense air humidity were the primary causes of the storm event.

The extreme storm event contained two phases: 1) a strong convective rain that occurred in the warm sector ahead of the cold front and 2) a dominant frontal rain after the arrival of the cold front (Guo et al, 2015). Those showed the reasons why the parameterizations of KF, WSM6 and MYJ were chosen for WRF rainfall prediction. KF has strong ability in simulating the low-level jet and the upward transportation of vapour (Kain, 2004). WSM6 contains six water substance variables, which can realistically identify rainfall formation (Kim et al, 2013). MYJ is more suitable for the simulation of the convection

system (Janjić, 1994). The 24 h accumulated rainfall with 172 mm led to the high peak flow (2580 m³ s⁻¹) in the Zijingguan

catchment. The observed areal rainfall in the Zijingguan catchment can be calculated using the Thiessen polygon method with the rainfall data from the 11 rain gauges, which were chosen as the ground truth to evaluate the WRF outputs. The forecasted areal rainfall was calculated by averaging values of the grid cells those have more than 50% area located inside the Zijingguan catchment. The 11 rain gauges with the Thiessen polygons and the model grid cells (3 km × 3km) were

shown in Fig. 1.

### 3.2 GTS data

Surface weather station, ship, buoy, pilot balloon, sonde, aircraft and satellite observations from the GTS can be processed using the OBSPROC observation preprocessor before being assimilated using WRF-3DVar. A shell script was compiled to transform the decoded data to the suitable LITTLE_R format, which can be directly used by WRF-3DVar for data

assimilation. Wide coverage in horizontal direction and high levels in vertical direction are the main characteristic of the GTS data, although the spatial density of the observations was low, and the time interval of the GTS data was 6 h. Therefore, the GTS data are usually used to improve the prediction of large-scale flows. In this study, five GTS datasets, including SOUND, SYNOP, PILOT, AIREP and METAR, were assimilated in the WRF model. The SOUND and SYNOP data took the majority of the GTS data. Detailed descriptions of the datasets were shown in Table 1. According to Fig. 2, the

observations covered by the outer domain were mostly located on land and only a few were on the ocean. The data located on land were distributed evenly.

**[Table 1 and Figure 2]**

The quality control of the GTS data is implemented in WRF-3DVar by defining the observation error covariance. The default US Air Force (AFWA) OBS error file is used in this study, which defines the instrumental and sensor errors for

various air, water and surface observation types as well as satellite retrievals.

### 3.3 Weather radar data

An S-band Doppler weather radar is located in Shijiazhuang, the capital of Hebei province. It is approximately 100 km from the Zijingguan catchment and covers a radius of 250 km. The study area can be completely covered by the radar. The radar cycles include 9 different scan elevations every 0.1 h over the course of a day. Figure 2 showed the relative positions of the

WRF domains and the radar scan area. To make the atmospheric motions more stable and reduce the nonlinearity in the outer domain, radar data were only assimilated in Domain 2 in this study.

The S-band radar belongs to the newest generation weather radar network of China (CINRAD/SC), the quality control of which is supported by China Integrated Meteorological Information Service System (CIMISS) of China Meteorological Administration. The potential error sources, such as ground clutter, radial interference echo, speckles and other artefacts,

were removed through the quality control procedure (Tong and Xue, 2005). The rainfall observations from rain gauges and weather radar were also compared to check the quality of the radar data. The following Z-R relationship was used to convert the radar reflectivity into rainfall rates (Hunter, 1996):

$$Z = 300 \times R^{1.4} \tag{2}$$

where Z is the radar reflectivity in $mm^6$ $m^{-3}$ and R is the rainfall rate in mm $h^{-1}$. Figure 3 showed the time series bars and the spatial distributions of the 24 h accumulated rainfall. The accumulated areal rainfall was 160.48 mm observed from the weather radar and 172.17 mm from the rain gauges. Although the accumulated rainfall was slightly underestimated by the

weather radar, the spatial distribution of the accumulation was quite consistent with the rain gauges. The temporal variation of the catchment areal rainfall also showed a consistent trend with the rain gauge observations. Therefore, the assimilation of the weather radar data is expected to have positive effect in improving the WRF forecasting results.

**[Figure 3]**

### 3.4 Mode configurations

To explore the effects of data assimilation using WRF-3DVar for rainfall prediction in the study area, 11 modes were designed based on the different combinations of the available GTS data, radar reflectivities and radial velocities in the two nested domains, as shown in Table 2. The improvements in the rainfall forecasts using data assimilation with the 11 modes were examined in this study and are shown in Table 2. Besides the different data assimilation combinations, the 11 modes have the same settings of the WRF model and the WRF-3DVar. The purpose of the mode design was to find the most

effective way to assimilate the weather radar and traditional meteorological data for improving the WRF rainfall forecasts.

**[Table 2]**

### 3.5 Cycling the WRF-3DVar runs

WRF-3Dvar needs to be run in the cycling mode to continuously assimilate real-time observations during storm events. According to the tests of different spin-up periods (6 h, 12 h and 24 h), which were commonly used in the WRF model

(Givati et al, 2012; Pan et al, 2017; Jr et al, 2016), the spin-up periods had little influence on rainfall prediction of the extreme storm event in this study. Considering the calculation efficiency, the 6 hour spin-up period was chosen. Figure 4 shows the start and end times of the storm event that occurred on 21 July 2012 and the time bars of the cycling WRF-3DVar runs. The storm event began at 03:00 on 21 July 2012 and lasted for 24 h. Data assimilation began at 00:00 on 21 July 2012 and ended at 00:00 on 22 July 2012, with a time interval of 6 h. Thus, the data assimilation took place on four occasions

(21/07/2012 00:00, 06:00, 12:00, 18:00 and 00:00). A 6 h spin-up period was executed by run1 and the first-guess files generated in run1 were used for run2. As time progressed, run3, run4, run5 and run6 were initiated at the corresponding times with the first-guess files generated by the previous runs. In the six runs, only run1 was the original WRF run without assimilating observations, which can be treated as a benchmark to evaluate the improvements using data assimilation.

**[Figure 4]**

# 4 Results

## 4.1 Evaluation on the storm process improvements

Figure 5 showed the forecasted rainfall process of the storm event on 21 July 2012 for the different data assimilation modes. To show the influence of the six runs on the rainfall forecast, all of the WRF-3DVar runs were shown in Fig. 5. The total run time was 36 h, which was longer than the duration (24 h) of the storm event. In Fig. 5, the gray area indicated the duration of the storm event. The black solid line indicated the ground truth of the catchment areal rainfall, which was calculated from the rain gauge observations using the Thiessen polygon method. The six runs were all shown by combining the solid and dashed lines. The solid line segment at the beginning of each run represented a new data assimilation run that generated the most recently updated forecasts. After 6 h, which marked the beginning of the next data assimilation run, the previous run was shown as a dashed line, indicating that the results were no longer the latest. For a given run, the solid and dashed line segments were the same color. The purpose of the data assimilation is to improve the accuracy of the rainfall forecast, and the presumed trends in the cumulative rainfall from the cycling WRF-3DVar runs should therefore increasingly approach the black solid line. In reality, different modes showed different data assimilation effects, and some of them were distinctly different from the presumed trends. Comparing the red and black hyetographs, the original forecast of the areal rainfall accumulation was significantly lower than the ground truth. This finding indicates that the WRF model was unable to forecast the storm event accurately without data assimilation, and the forecast errors may lead to poor runoff forecasts due to error accumulation and magnification during the rainfall-runoff transformation process.

Comparing the first three subfigures of Fig. 5, a significant improvement in the accumulated areal rainfall was found in Mode 1 by assimilating the radar reflectivity in Domain 2. The results of Modes 2 and 3 were unstable in the different cycling WRF-3DVar runs, which was a result of the assimilation of the radar velocity data. For Modes 2 and 3, the accumulated areal rainfall forecasts for the first 6 h of run3 were less than for Mode 1, and the assimilation results of run4 were also unsatisfactory. The forecasted rainfall for Modes 2 and 3 was even less than the original run (run1), which indicated that the assimilation of radial velocity observations at the time 21/07/2012 06:00 and 12:00 was unable to help trigger the main storm process.

For Modes 4 and 5 shown in Fig. 5d and 5e, assimilating the GTS data improved the rainfall forecasts, and only assimilating the GTS data in Domain 1 (Mode 4) was a little better than assimilating the GTS data in both nested domains (Mode 5). The accumulated areal rainfall forecast for Mode 4 was only slightly better than that of Mode 1, while the rainfall processes were very different. For Mode 1, the greatest rainfall amounts were obtained in run3, whereas the rainfall amounts of run4 had the largest proportion for Mode 4. Different types of data assimilation can not only affect the rainfall amounts but also the rainfall process.

In order to investigate the impacts of assimilating GTS data on the radar data assimilation effects and the rainfall forecasts, Mode 6, 7 and 8 (assimilating the GTS data in Domain 1 as well as the radar data in Domain 2) were compared with Mode 1, 2 and 3 (only assimilating the radar data in Domain 2). For the six modes, the rainfall processes were relatively similar for

run1, run2, run5 and run6, whereas the forecasts of run3 and run4 were very different. The accumulated rainfall for run3 in Mode 1 was slightly higher than in Mode 6, whereas the result was opposite for run4. In comparison with Modes 2 and 3, Modes 7 and 8 had larger accumulated rainfall totals in run3 and run4, respectively. This finding indicated that the assimilation of the GTS data in Domain 1 could have affected the rainfall forecast when assimilating radar data in Domain 2, which resulted in further improved forecasts that agreed better with the observed rainfall.

Modes 9, 10 and 11 were designed to explore the influence of the assimilation of the GTS data in Domain 2 on Modes 6, 7 and 8. The accumulated rainfall for run4 in Mode 9 was less than in Mode 6, which indicated that assimilating the GTS data in Domain 2 at 21/07/2012 12:00 may have changed the atmospheric state and water vapor transport when the GTS data were also assimilated in Domain 1 and the radar reflectivity data were assimilated in Domain 2 at the same time. Both run3 and run4 of Mode 10 had much less forecasted rainfall than Mode 7, and the rainfall forecast was the worst among the 11 modes. This result showed that assimilating the GTS data in Domain 2 may have reduced the improvement in the rainfall forecast when the GTS data were also assimilated in Domain 1 and the radial velocity data were assimilated in Domain 2. Unlike Modes 9 and 10, Run4 of Mode 11 predicted more rainfall than the other modes and provided the largest contribution to the 24 h accumulated areal rainfall.

It can be found in Fig. 5 that the predicted storms always start and end around 6-h earlier than the observations. Besides the errors in the boundary conditions, it is found that the assimilation of the water vapor information (contained in the radar reflectivity and the GTS data) can make the rain in the initial fields form and fall to the earth more quickly (Georgakakos, 2000; Sun, 2005; Sun et al, 2016). Considering the error is consistent, an error prediction model could be built in further studies, and the assimilation of the latent heat may also be helpful in correcting the starting and ending time of the forecasted rainfall process (Stephan et al, 2010; Schraff et al, 2016).

**[Figure 5]**

## 4.2 Evaluation on the 24 h accumulated areal rainfall

To more quantitatively evaluate the 11 data assimilation modes, the accumulated areal rainfall in the Zijingguan catchment was calculated for the 24 h duration of the storm event on 21 July 2012. The rain gauge observations, WRF model forecasts, and relative errors between them were shown in Table 3. For all 11 modes, the first six hours of the runs (solid line segments in Fig. 5), which covered the duration of the storm, were used to calculate the 24 h accumulated areal rainfall.

**[Table 3]**

The WRF model forecasts without data assimilation were too poor to be used for hydrological forecasting. The forecasted rainfall accumulation was only 95.54 mm, which was much lower than the rain gauge observations. Data assimilation was used to improve the rainfall forecasts, although the forecast results worsened in Modes 2, 3 and 10, and all of the relative errors exceeded 50%. Interestingly, all three modes assimilated radar velocity data. For the other eight modes, the rainfall forecasts were improved at different levels. Mode 11 performed the best because the forecast (165.68 mm) was closest to the

rain gauge observation (172.17 mm), and the relative error was only -3.77%. Although Mode 5 was improved by assimilating the GTS data in both nested domains, the effect was the least profound.

Comparing Modes 1, 2, 3, and 4, which all assimilated the observations in a single domain, the lower relative errors were found in Modes 1 (-30.93%) and 4 (-25.06%), whereas the highest error was found in Mode 2 (-54.90%). This finding indicated that assimilating radar reflectivity and GTS data can provide good results, while assimilating radar velocity made the rainfall forecast unsatisfactory. The observation errors in the radial velocity may have been the main factor that led to the poorest performance (Abhilash et al, 2012). Another reason is that assimilating radial velocity can only change the dynamic field, which changes quickly for small-scale regions, and 6 h may have been too long for the assimilation time interval (Lin et al, 2011).

Modes 6, 7 and 8 were formed by assimilating the GTS data in Domain 1 based on Modes 1, 2 and 3. The results showed that the relative errors decreased significantly when the GTS data were assimilated in Domain 1. The forecasted rainfall amounts for Modes 6, 7 and 8 were very similar, and the relative errors were all approximately -20%. This finding indicated that assimilating the GTS data over a large area (Domain 1) had large positive effects on the rainfall forecast using the WRF model.

Compared with Modes 6, 7 and 8, Modes 9, 10 and 11 also assimilated the GTS data in Domain 2 at the same time. The difference between the forecasted rainfall totals for Modes 6 and 9 was not significant, although Mode 9 performed worse than Mode 6. However, the relative error for Mode 10 was -58.74%, which was much higher than of Mode 7. This finding indicated that assimilating the GTS data over a small area (Domain 2) may have increased the forecast error, whereas the forecasted rainfall may not have been greatly influenced when the radar reflectivity was assimilated in Domain 2 at the same time.

## 4.3 Evaluation on the spatial and temporal distributions of rainfall

Beside the accumulative amount, the rainfall variations in both space and time play a crucial role in the formation of the runoff process. Therefore, in the study the spatial and temporal distributions of the forecasted rainfall were evaluated by the root mean square error (RMSE) and the results were shown in Table 4. The rainfall accumulations at different rain gauges and the catchment average rainfall at different time steps are respectively used to calculate the RMSE indices in the spatial and the temporal dimensions. Detailed equations can be found in Tian et al. (2017a). The spatial and temporal distributions of the forecasted 24-h rainfall from the 11 data assimilation modes were shown in Fig. 6 and 7 respectively. It can be observed that for most of the assimilation modes the spatial and temporal distributions were improved in varying degrees after data assimilation. Mode 6 had the lowest RMSE (0.316) in the spatial dimension and Mode 7 showed the lowest RMSE (0.582) in the temporal dimension.

When the radar data are solely assimilated with the radial velocity being involved, Mode 2 and Mode 3 showed the worst distribution results among the 11 modes, and no improvement was found in both spatial and temporal dimensions. Mode 1 performed much better than Mode 2 and Mode 3 for the spatial distributions of the rainfall forecasts while the RMSEs of the

temporal distributions for Mode 1, 2 and 3 did not have much difference. Mode 4 and 5 showed improvements to certain extends in both spatial and temporal distributions after data assimilation.

**[Table 4, Figure 6 and 7]**

For the temporal distributions of the rainfall forecasts, the RMSEs of Mode 1, 2 and 3 were above 0.90, while the RMSEs of Mode 6, 7 and 8 were much lower, between 0.58~0.71. For the spatial distributions, the RMSEs were above 0.80 for Mode 2 and 3 but less than 0.40 for Mode 7 and 8, and the RMSE was also reduced from 0.456 for Mode 1 to 0.316 for Mode 6. Compared with Mode 1, 2 and 3, the rainfall distributions of Mode 6, 7 and 8 had significant improvements by adding the GTS data assimilation in the outer domain, which also indicated that assimilating the GTS data over a large area can help the WRF model assimilate the radar data more effectively.

However, the rainfall distributions of Model 9, 10 and 11 were worse compared with Mode 6, 7 and 8, though Mode 9 performed a little better than Mode 6 in the temporal dimension and Mode 11 was better than the Mode 8 in the spatial dimension. Especially for Mode 10, the RMSEs for the spatial and temporal distributions of the rainfall forecasts were both higher than Mode 7. The results indicated that assimilating the GTS data in the inner domain may have a negative impact on the assimilation of the radar data regarding the distributions of the forecasted rainfall.

## 4.4 Influence of the number of assimilated observations

To further explore the assimilation techniques and the causes of the assimilation effects, the number of assimilated observations for each of the 11 modes is shown in Table 5.

**[Table 5]**

Mode 3 can be regarded as a combination of Modes 1 and 2. The number of assimilated observations in Mode 3 was less than the sum of the number of assimilated observations in Modes 1 and 2 at each time. Therefore, there was a conflict between the radar reflectivity and radial velocity, and some of the radar data were not assimilated in the WRF model when the two types of radar data were assimilated in the same domain at the same time. However, Mode 3 performed worse than Mode 1, which indicated that assimilating the radial velocity data in Domain 2 had a negative effect on the forecasted rainfall.

Through a comparative analysis of Modes 1 and 6, Modes 2 and 7, and Modes 3 and 8, Modes 6, 7 and 8 assimilated fewer observations than Modes 1, 2 and 3, although the assimilation effects were improved. This finding indicated that the data assimilation approach in Modes 6, 7 and 8 was more rational. The assimilation of the GTS data in Domain 1 helped eliminate the unreasonable radar data assimilated in Modes 1, 2 and 3 and improved the radar data assimilation results.

The comparisons were also made between Modes 6 and 9, Modes 7 and 10, and Modes 8 and 11. For Modes 9, 10 and 11, the number of assimilated observations in Domain 2 was more than for Modes 6, 7 and 8, which meant that the GTS data were assimilated by the WRF model in Domain 2. However, the assimilation results indicated that more assimilated data did not mean better assimilation effects.

**5 Discussion**

Among the 11 data assimilation modes, assimilating radar velocity always led to poorer results than the original rainfall forecasts. It should be noted that the assimilation of the radar radial velocity cannot directly influence the physical process of the rainfall formation, although the assimilation can change the wind field and affect the water vapor transport. If the assimilated radar velocity cannot improve the atmospheric circulation predictions, the water vapor field may not be improved, which thus leads to unimproved rainfall forecasts (Pan et al, 2012; Dong and Xue, 2013). In reality, the accuracy of the radial velocity observations depends on the atmospheric refractive index, which is affected by the air density and the water vapor content (Montmerle and Faccani, 2010; Maiello et al, 2014). Unfortunately, both the air density and the water vapor content are quite variable, especially on rainy days (Abdalla and Cavaleri, 2002). Therefore, the spatial observation errors in the radial velocity retrievals are unavoidable and might be the main factor that leads to poorer performance of the NWP model than that achieved without data assimilation (Abhilash et al, 2012). Due to the ability of frequent adjustment for the atmospheric motions, decreasing the assimilation time interval may reduce the risk of over correction (Xiao et al, 2005). However, the added information may involve more observation errors, which may increase the nonlinearity of the atmosphere and lead to a difficult model convergence. The weather radar in this study had a temporal resolution of 0.1 h, whereas the assimilation time interval was 6 h; therefore, there is still room to increase the data assimilation frequency, although there is a trade-off between model performance and the operating efficiency.

On the contrary, the assimilation of the radar reflectivity always had a positive effect on the forecasted rainfall, and the model performance was relatively stable. The data assimilation modes which involved the radar reflectivity always performed better than the others. The main reason is that the radar reflectivity contain information directly related to the precipitation hydrometeors. According to the Eq. (2), the assimilation of radar reflectivity is a correction to the humidity field in essence, which directly influences the formation process of the precipitation. Additionally, after strict quality control the radar reflectivity assimilated in this study showed consistent trends with the gauge observations in both spatial and temporal distributions (as shown by Fig. 3), which helped result in more effective assimilation results than the radial velocity. The assimilation of the GTS data played a subsidiary role to the radar reflectivity in improving the rainfall forecasts. A combination of the radar reflectivity together with the GTS data always resulted in better results that assimilating the radar data alone. Although the spatial density of the GTS data is relatively low compared to the radar data, it contains observations of various atmospheric states. As shown by Fig. 2, the GTS data located on land were distributed evenly, which helped improve the stability of the WRF model during data assimilation (Carrassi et al, 2008). SOUND and SYNOP datasets took the majority of the GTS data (Table 1), which means that the observations from the surface-based stations and the upper-air observatories have the most contributions to the improvement of the rainfall prediction. Pressure, temperature, humidity and wind reports from the surface and upper air are contained in SOUND and SYNOP datasets. The assimilation of these meteorological elements can directly correct the initial and lateral boundary conditions through the wide horizontal coverage and vertical levels (Tu et al, 2017). Figure 8 compares the improvement of the initial conditions of Domain 2 after data

assimilation in Mode 1 and Mode 6. The increment distributions of wind, humidity and pressure when the radar reflectivity was solely assimilated in the inner domain (Mode 1) were shown in Fig. 8a, whereas those when assimilating the radar reflectivity in the inner domain as well as the GTS data in outer domain (Mode 6) were shown in Fig. 8b.

**[Figure 8]**

It can be seen that the changes of the wind increments was more significant in Fig. 8b than Fig 8a in the inner domain of the WRF model. Though the changes of the increment distributions for humidity and pressure were not obvious for the two data assimilation modes, the major difference gather toward the southeast of the domain area, which assisted in the occurrence of the heavy rainfall in the downstream of the study catchment. This may to some extend explain why the rainfall accumulations and the spatial and temporal distributions can be improved from Mode 1 to Mode 6, by adding the GTS data

in the outer domain. The involvement of the GTS data in the outer domain helped improve the atmospheric state at a relatively large scale, which induced a positive effect on the rainfall forecasts in the inner domain with the two-way nesting mechanism. Therefore, in this study the assimilation of GTS data together with the radar data performs the best among different assimilation modes.

In order to further verify the findings for assimilating different sources of data, another two storm events with relatively

moderate rainfall intensity (around 50mm for 24 h accumulation) were selected from an analogue catchment to be tested with four simplified assimilation modes (Mode 1, 3, 4, 6). The analogue catchment locates in the southern reach of the Daqing river basin with a drainage area of 2210 km$^2$. Table 6 showed the details of the storm events and the rainfall forecasting results from different data assimilation modes. The cumulative hyetographs of the forecasted areal rainfall were illustrated in Fig. 9. The results were in accordance with the conclusions from the extreme event in Zijingguan catchment.

Assimilating radar reflectivity together with radial velocity (Mode 3) cannot guarantee improved results than only assimilating radar reflectivity (Mode 1). The GTS data alone can also generate as good results (Mode 4) as the radar reflectivity. When the radar reflectivity is assimilated in the inner domain, the involvement of the GTS data in the outer domain (Mode 6) further helped produce better results.

**[Figure 9 and Table 6]**

Before the NCEP driven data was used in this study, ECMWF was also tested for the data assimilation with the same storm event. Although the rainfall forecasts showed some differences based on the boundary conditions from the two centres, the improvement patterns from different data assimilation modes were quite similar. It should be noted that the initial and lateral boundary conditions do have some potential impact on the rainfall forecasts results. More studies should be carried out to verify the effects of data assimilations using different driven data, such as CMA. The ultimate goal for the application of the

numerical rainfall prediction is to make flow forecasts at the catchment outlet. Errors in the forecasted rainfall process and the accumulative amount can result in divergent flood peak time and peak stage of the flood (Shih et al, 2014). Therefore, data assimilation is an important tool in improving the forecasted rainfall as well as the flow. Yucel et al. (2015) assimilated conventional meteorological observations to improve the rainfall prediction, meanwhile the mean runoff error was reduced by 14.7% with data assimilation in the Black Sea Region. The peak discharge error was also reduced from 50% to 14% when

the radar data were assimilated in a hydro-meteorological model built in the Dese river catchment (Rossa et al., 2010). In the further study, the coupled atmospheric-hydrological model with data assimilation will be built and flood forecasts from the coupling system will be examined to evaluate the effect of data assimilation on flood forecasting.

## 6 Conclusion

This study explored the effects of data assimilation using WRF-3DVar for the improvement of rainfall forecasting in the Beijing-Tianjin-Hebei region of Northern China. Two nested domains were employed, and the GFS data were used to drive the WRF model. A storm event that occurred over the Beijing-Tianjin-Hebei region on 21 July 2012 was selected, considering the widespread attention it received in China due to the high intensity, wide coverage of the rainfall process and the significant losses caused by the following flood. The rainfall accumulation during the storm in a mountainous catchment
named Zijingguan with a drainage area of 1760 km$^2$ was set as the forecasting target. Three types of observations, i.e., GTS data, radar reflectivity and radial velocity, were used to investigate the potential improvements on WRF rainfall forecasts through data assimilation. Eleven data assimilation modes were designed based on different combinations of the three types of observations in the two nested domains.

Contrastive analyses of the rainfall forecasts from the 11 data assimilation modes were carried out from three aspects: the
15 rainfall evolution process, the accumulated amount and the number of the observations assimilated. Four main conclusions can be drawn: 1) when the radar data was assimilated alone, the assimilation of radar reflectivity performed better than radial velocity in improving the rainfall forecasts, and the assimilation of both two types of the radar data generated poorer results than only assimilating the radar reflectivity; 2) assimilating the GTS data over a large area can help the WRF model assimilate the radar data effectively, and the involvement of the GTS data in the outer domain when radar data are
20 assimilated in the inner domain resulted in better forecasts than only assimilating the radar data; 3) assimilating more observations does not guarantee further improvement, on the other hand, the effective information contained in the assimilated data is of more importance than the data quantity. In this study, according to the results of 24-h accumulated areal rainfall prediction, spatiotemporal distributions of the predicted rainfall and the number of assimilated observations, the combination of radar reflectivity and GTS data resulted in the best improvements of the rainfall forecasts. Considering the
25 poor performance of radial velocity in other assimilation modes and the reasons explained in the discussion, the radial velocity data should be carefully used with a very strict quality control process. Further research considering various geographical and meteorological case studies should be carried out to further verify the conclusions of this study.

**Author Contributions**

All the authors contributed to the conception and the development of this manuscript. Jia Liu and Jiyang Tian contributed to the calculations and the analyses. Denghua Yan and Fuliang Yu assisted in the experiment design and the assimilation methods. Chuanzhe Li and Feifei Shen helped with the figure production and the manuscript writing.

**Acknowledgements**

This study was supported by the National Key R&D Program (2016YFA0601503, 2017YFC1502405), the Hebei Province Water Scientific Research Project (2015-16), the Major Science and Technology Program for Water Pollution Control and Treatment (2018ZX07110001), and the IWHR Research & Development Support Program (WR0145B732017).

The authors especially appreciate the helpful comments and constructive suggestions from the editor, Dr. Florian Pappenberger, and the two reviewers, Dr. Yunqing Xuan and an anonymous reviewer.

**Competing interests:** The authors declare that they have no conflict of interest.

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

**Table captions**

Table 1: Descriptions of the GTS datasets assimilated in the study.

Table 2: Designed assimilation modes with different data combinations.

Table 3: Observed and forecasted 24 h accumulated areal rainfall from different data assimilation modes.

Table 4: RMSEs for the spatial and temporal distributions of the forecasted rainfall from different data assimilation modes.

Table 5: Total number of assimilated observations in the two WRF nested domains.

Table 6: The two storm events of the analogue catchment and the relative errors of the forecasted rainfall from different data assimilation modes.

**Figure captions**

Figure 1: The location of the Zijingguan catchment with (a) the 11 rain gauges and (b) the 3×3 km grid cells.

Figure 2: Locations of the radar scan area, the GTS data, the study catchments and the two nested domains.

Figure 3: Rainfall observations from the rain gauges and the weather radar: (a) hyetograph of the hourly catchment areal rainfall; (b) spatial distribution of the 24 h rainfall accumulation from the rain gauges; (c) spatial distribution of the 24 h rainfall accumulation from the radar.

Figure 4: The time bars of the cycling WRF-3DVar runs.

Figure 5: Cumulative hyetographs for the 11 data assimilation modes.

Figure 6: Spatial distributions of the forecasted 24-h rainfall accumulations from different data assimilation modes.

Figure 7: Hyetographs of the observed and forecasted rainfall from different data assimilation modes.

Figure 8: The increment distributions of wind, humidity and pressure in Domain2 at 850hPa for (a) assimilating only radar reflectivity and (b) assimilating both radar reflectivity and GTS data.

Figure 9: Cumulative hyetographs of the two storm events in the analogue catchments with different data assimilation modes.

**Table 1.** Descriptions of the GTS datasets assimilated in the study.

| Dataset | Descriptions | Number of observations |
|---|---|---|
| SOUND | Upper-level pressure, temperature, humidity and wind report from a fixed or mobile land station, a sea station or a sonde released by carrier balloons or aircraft. | 2718 |
| SYNOP | Report of surface observation from a fixed or mobile land station. | 4217 |
| PILOT | Upper-wind report from a fixed or mobile land station or a sea station. | 733 |
| AIREP | Aircraft weather report. | 201 |
| METAR | Aerodrome routine or special meteorological report. | 612 |

**Table 2.** Designed assimilation modes with different data combinations.

| Modes | Assimilated data | |
| --- | --- | --- |
| | Domain 1 | Domain 2 |
| 1 | / | radar reflectivity |
| 2 | / | radial velocity |
| 3 | / | radar reflectivity and radial velocity |
| 4 | GTS data | / |
| 5 | GTS data | GTS data |
| 6 | GTS data | radar reflectivity |
| 7 | GTS data | radial velocity |
| 8 | GTS data | radar reflectivity and radial velocity |
| 9 | GTS data | GTS data and radar reflectivity |
| 10 | GTS data | GTS data and radial velocity |
| 11 | GTS data | GTS data, radar reflectivity, and radial velocity |

**Table 3.** Observed and forecasted 24 h accumulated areal rainfall from different data assimilation modes.

| Modes | Rain gauge (mm) | WRF model (mm) | Relative error (%) |
|---|---|---|---|
| No data assimilation | 172.17 | 95.54 | -44.51 |
| 1 | 172.17 | 118.92 | -30.93 |
| 2 | 172.17 | 77.65 | -54.90 |
| 3 | 172.17 | 79.76 | -53.67 |
| 4 | 172.17 | 129.02 | -25.06 |
| 5 | 172.17 | 111.71 | -35.11 |
| 6 | 172.17 | 136.37 | -20.79 |
| 7 | 172.17 | 132.89 | -22.82 |
| 8 | 172.17 | 132.89 | -22.81 |
| 9 | 172.17 | 124.74 | -27.55 |
| 10 | 172.17 | 71.04 | -58.74 |
| 11 | 172.17 | 165.68 | -3.77 |

**Table 4.** RMSEs for the spatial and temporal distributions of the forecasted rainfall from different data assimilation modes.

| Modes | Spatial dimension | Temporal dimension |
|---|---|---|
| No data assimilation | 0.456 | 0.902 |
| 1 | 0.421 | 0.914 |
| 2 | 0.834 | 0.921 |
| 3 | 0.826 | 0.934 |
| 4 | 0.337 | 0.692 |
| 5 | 0.539 | 0.887 |
| 6 | 0.316 | 0.701 |
| 7 | 0.414 | 0.582 |
| 8 | 0.359 | 0.613 |
| 9 | 0.362 | 0.667 |
| 10 | 0.833 | 1.232 |
| 11 | 0.341 | 0.826 |

**Table 5.** Total number of assimilated observations in the two WRF nested domains.

| Modes | Domain | Number of assimilated observations | | | | |
|---|---|---|---|---|---|---|
| | | Time 1 | Time 2 | Time 3 | Time 4 | Time 5 |
| | | 21/07 00:00 | 21/07 06:00 | 21/07 12:00 | 21/07 18:00 | 22/07 00:00 |
| 1 | 1 | / | / | / | / | / |
| | 2 | 16158 | 19233 | 26032 | 11821 | 6239 |
| 2 | 1 | / | / | / | / | / |
| | 2 | 75014 | 107901 | 57095 | 60993 | 23508 |
| 3 | 1 | / | / | / | / | / |
| | 2 | 60550 | 84573 | 71148 | 45896 | 24924 |
| 4 | 1 | 2625 | 1758 | 2445 | 930 | 723 |
| | 2 | / | / | / | / | / |
| 5 | 1 | 2625 | 1758 | 2445 | 930 | 723 |
| | 2 | 596 | 170 | 304 | 145 | 106 |
| 6 | 1 | 2625 | 1758 | 2447 | 931 | 720 |
| | 2 | 6156 | 2905 | 14795 | 3795 | 4232 |
| 7 | 1 | 2625 | 1760 | 2450 | 934 | 721 |
| | 2 | 54394 | 14954 | 40227 | 52914 | 20984 |
| 8 | 1 | 2625 | 1760 | 2450 | 931 | 724 |
| | 2 | 60550 | 17806 | 72966 | 39142 | 23912 |
| 9 | 1 | 2625 | 1759 | 2444 | 937 | 723 |
| | 2 | 6752 | 3095 | 31711 | 42108 | 20194 |
| 10 | 1 | 2625 | 1758 | 2446 | 935 | 719 |
| | 2 | 54990 | 64247 | 78253 | 53209 | 24628 |
| 11 | 1 | 2625 | 1760 | 2449 | 941 | 725 |
| | 2 | 61146 | 17823 | 74014 | 41592 | 24293 |

**Table 6.** The two storm events of the analogue catchment and the relative errors of the forecasted rainfall from different data assimilation modes.

| Event | Duration | Rain gauge observation (mm) | WRF original run (%) | Assimilation results (%) | | | |
|---|---|---|---|---|---|---|---|
| | | | | Mode 1 | Mode 3 | Mode 4 | Mode 6 |
| a | 29/07/2007 20:00-30/07/2007 20:00 | 63.38 | -15.94 | -11.45 | -14.66 | -9.35 | -2.78 |
| b | 30/07/2012 10:00-31/07/2012 10:00 | 50.48 | -26.26 | -9.69 | -17.16 | -6.01 | -2.10 |

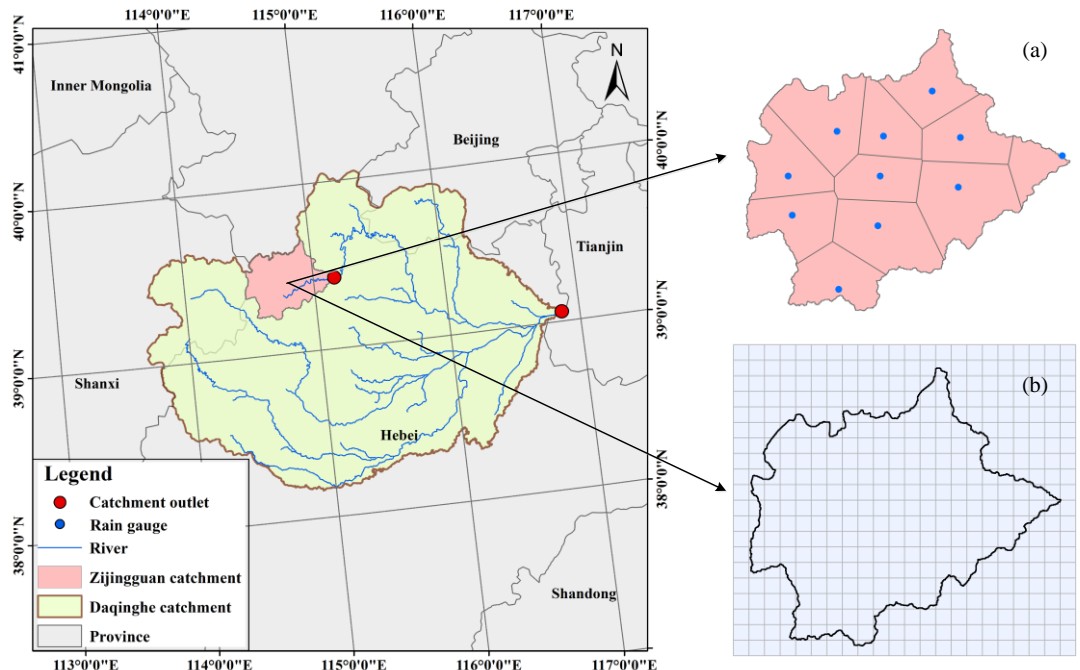

**Figure 1.** The location of the Zijingguan catchment with (a) the 11 rain gauges and (b) the 3×3 km grid cells.

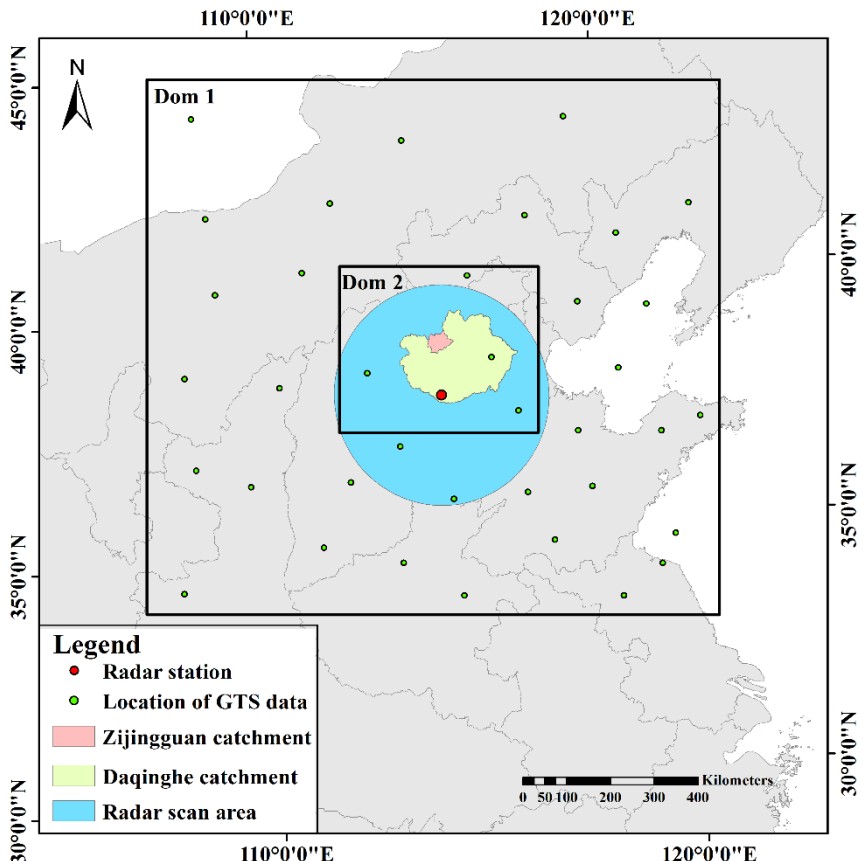

**Figure 2.** Locations of the radar scan area, the GTS data, the study catchments and the two nested domains.

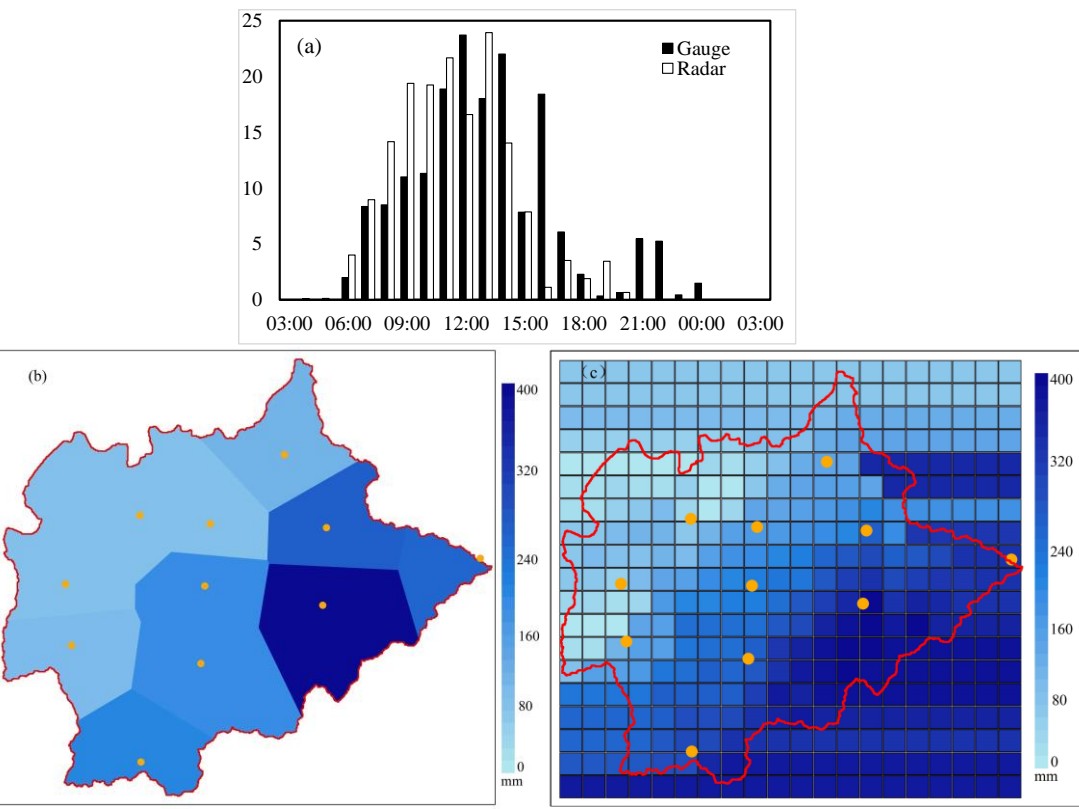

**Figure 3.** Rainfall observations from the rain gauges and the weather radar: (a) hyetograph of the hourly catchment areal rainfall; (b) spatial distribution of the 24 h rainfall accumulation from the rain gauges; (c) spatial distribution of the 24 h rainfall accumulation from the radar.

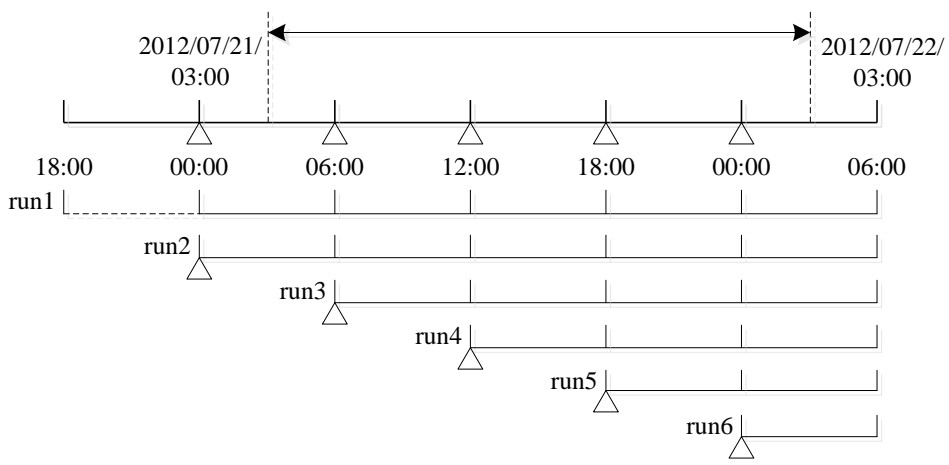

**Figure 4.** The time bars of the cycling WRF-3DVar runs.

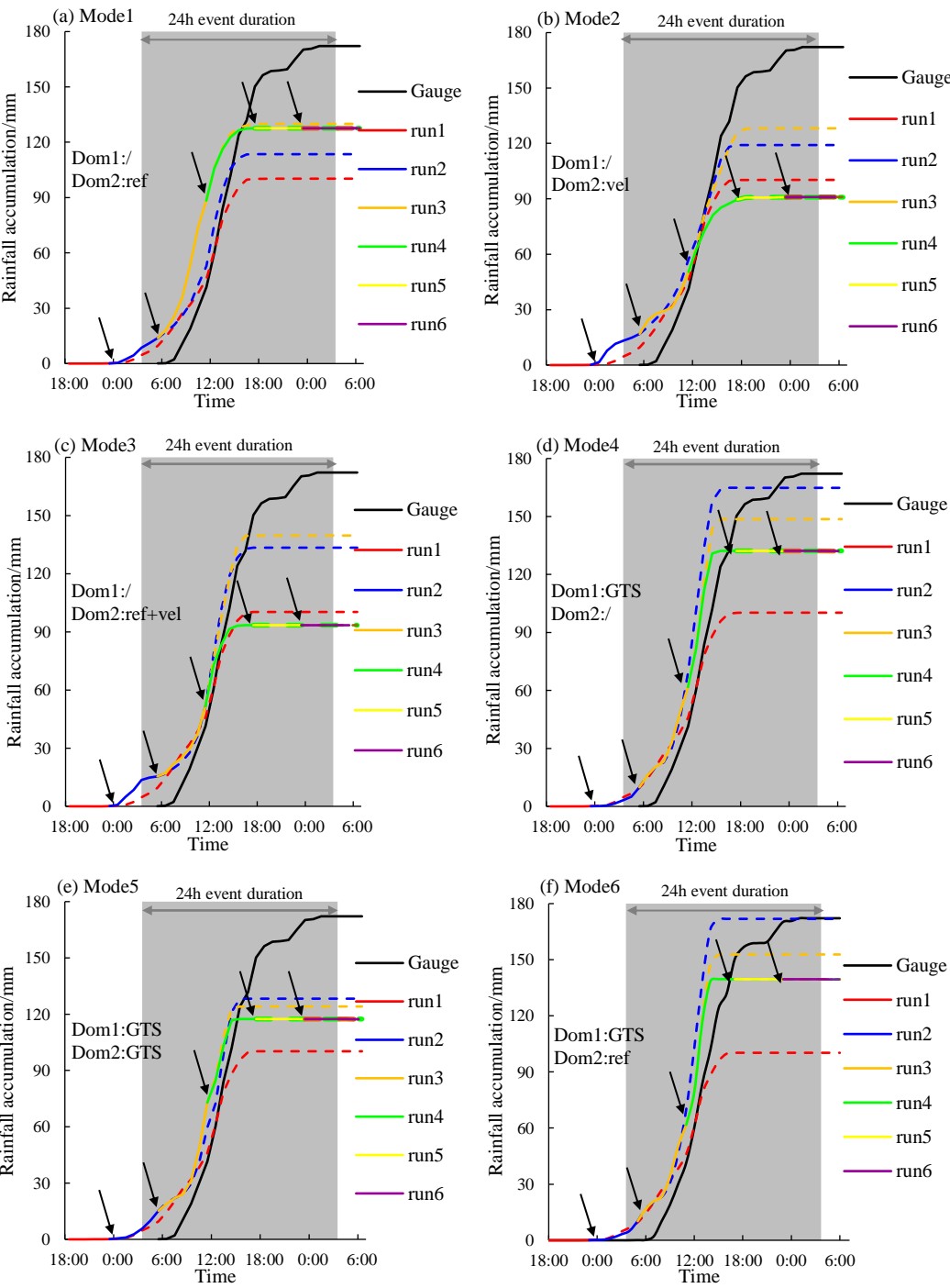

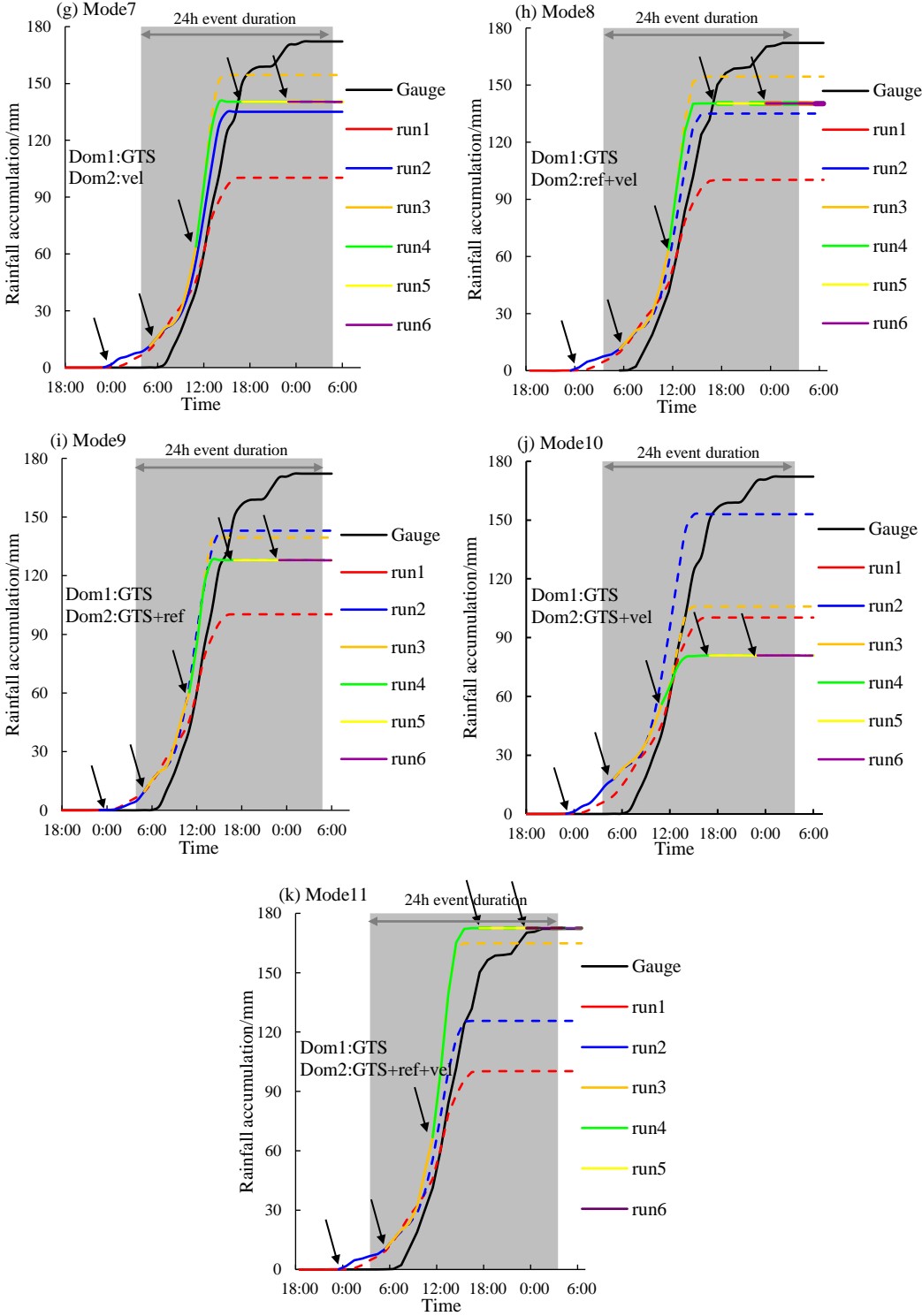

**Figure 5.** Cumulative hyetographs for the 11 data assimilation modes.

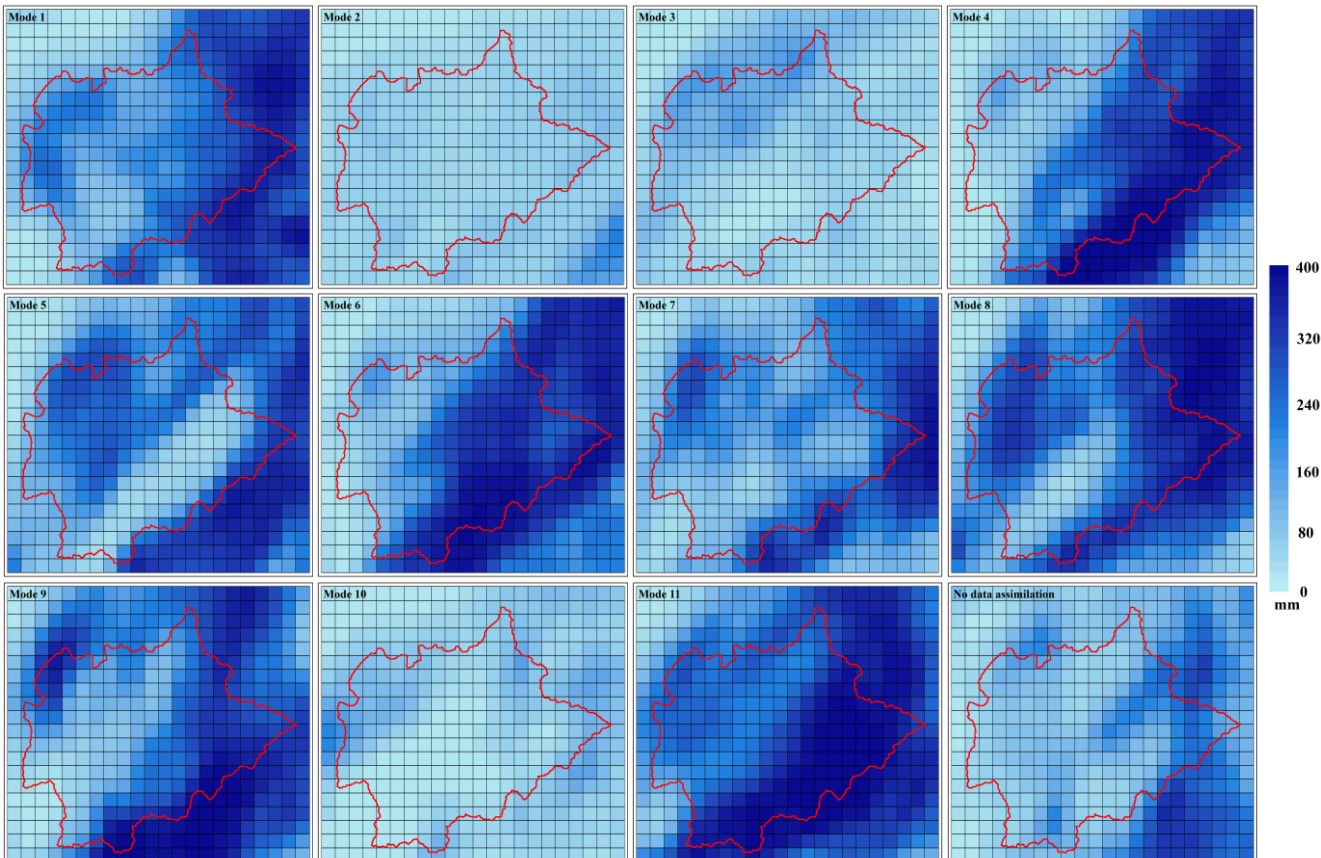

**Figure 6.** Spatial distributions of the forecasted 24-h rainfall accumulations from different data assimilation modes.

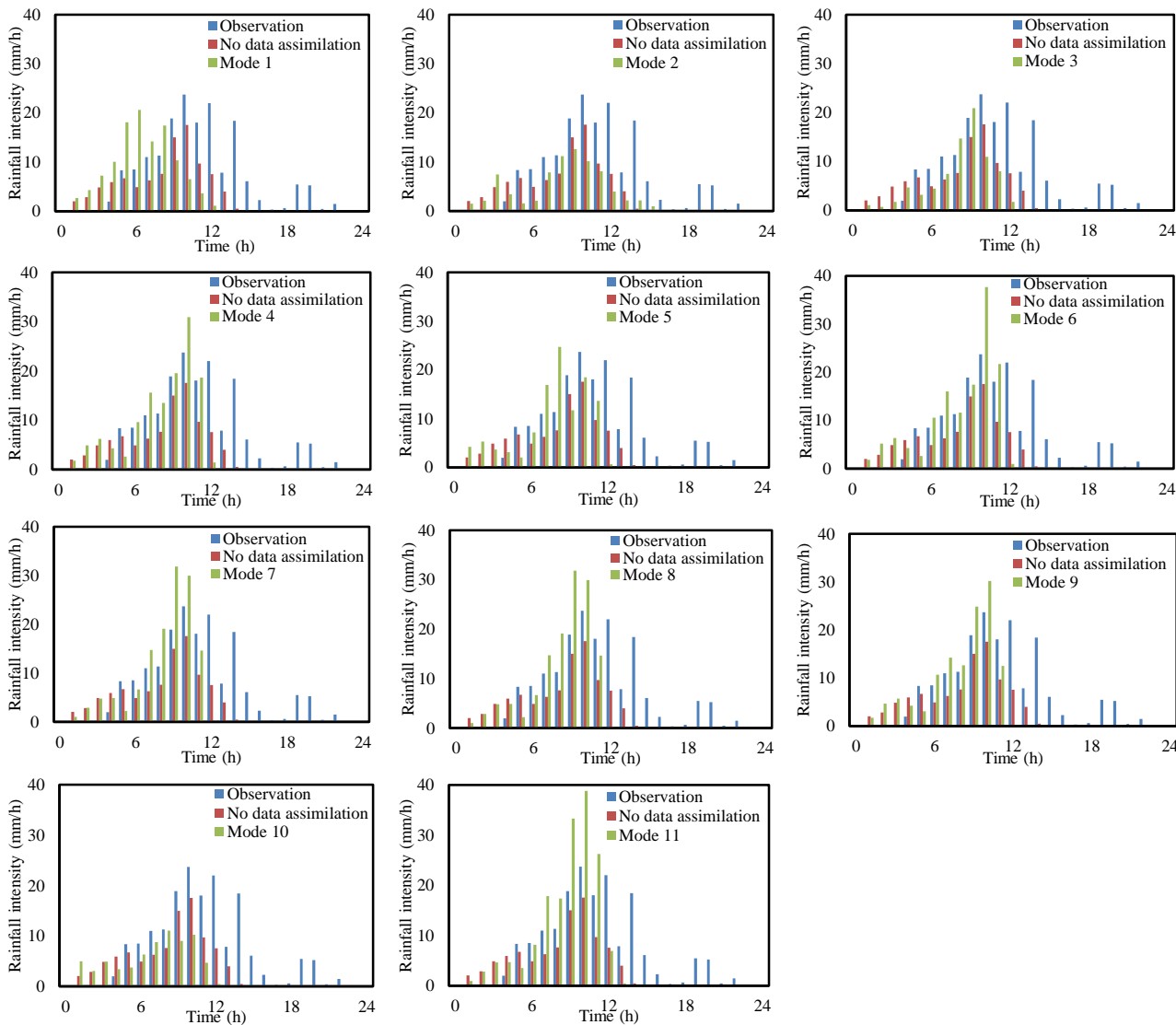

**Figure 7.** Hyetographs of the observed and forecasted rainfall from different data assimilation modes.

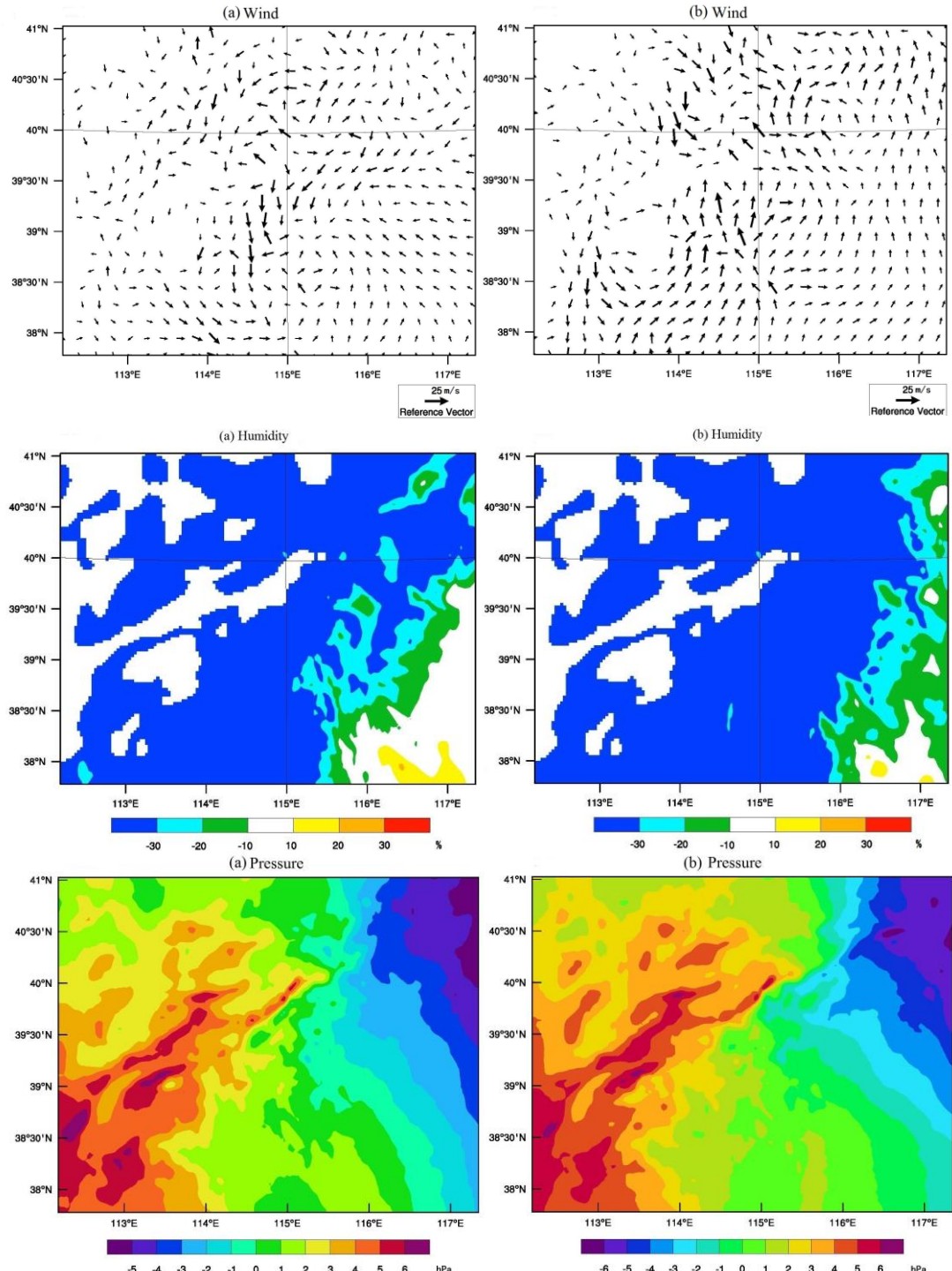

**Figure 8.** The increment distributions of wind, humidity and pressure in Domain2 at 850hPa for (a) assimilating only radar reflectivity and (b) assimilating both radar reflectivity and GTS data.

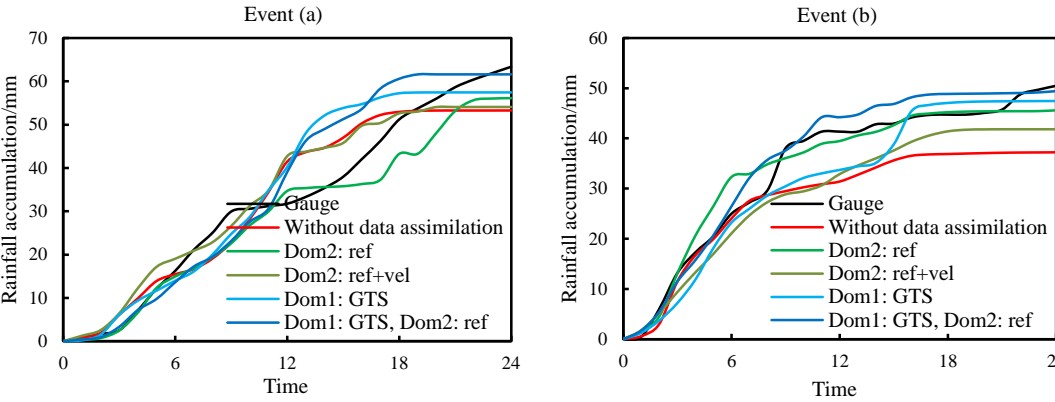

**Figure 9.** Cumulative hyetographs of the two storm events in the analogue catchments with different data assimilation modes.