# Peer review of "Evaluation of Doppler radar and GTS Data Assimilation for NWP Rainfall Prediction of an Extreme Summer Storm in Northern China: from the Hydrological Perspective"

_Hydrology and Earth System Sciences, 2017_

## Referee Comment (RC1) · Anonymous Referee #1 · 22 Jan 2018

This paper contains a comparison of eleven data assimilation modes, which are designed for assimilating different combinations of weather radar and GTS data in the two nested domains of the WRF model for rainfall prediction. It is always interested to see the efforts that try to integrate weather radar (providing local, short-term lead time but high spatiotemporal resolutions rainfall) with NWP model (regional, medium lead time but relatively poor-quality rainfall) as they are very complementary. Since the WRF model and the data assimilation methodology is increasingly applied in the hydrology and earth system areas, how to use the model and the data correctly and efficiently be-

comes an important issue that is worth discussing. The results of this paper are helpful for improving the rainfall prediction from the hydrological perspective and the content is of interest to the readers of the HESS journal. However, further improvements and clarifications are needed before the paper is acceptable. Detailed comments are listed below:

1) I agree with the authors that hydrologists are particularly concerned about the accuracy of the accumulative amount and the process of the predicted rainfall at the catchment scale. However, I did not observe any special configuration of WRF or data assimilation for this goal.

2) What factors drove your decision to have 40 vertical levels in WRF? Why not more?

3) The uncertainty of weather radar rainfall and GTS data, and their possible influences on the data assimilation process should be carefully specified.

4) The spin-up period may have influences on the rainfall simulation of the NWP. A 6 hour is used in this study, why not 12 h or other times? In fact, the authors should specify or at least discuss all the sense parameterizations, such as the domain design, downscaling ratios ect.

5) It is necessary to explain the causes of the heavy rainfall event in this study. The readers may be interested in the very convective system which affects the choices of the WRF parameterisation schemes.

6) The description of assimilation mechanism for the radar data is too simple. Please complement the model-derived observation operators that are adopted in the 3DVar method. This may explain why the assimilation of the radar reflectivity performs better than the radial velocity.

7) In general, this study describes the results of different data assimilation experiments and explains the reasons why the assimilation of the radar velocity always leads to worse results. It is wondered why the GTS data and the radar reflectivity can perform

better, and neither does the author give a plausible explanation why the assimilation of GTS data together with the radar data performs the best among the eleven assimilation modes. The revised manuscript needs to contain more deep analysis in the Discussion section.

8) The ultimate goal of the WRF applications in this study is for flood forecasts. It would be helpful to add references to explain how much the flood forecasts in general can be improved by the improvement of the rainfall accuracy. I also look forward to the follow-up study for the improvement of flood forecasts with data assimilation.

9) There are several typos and some cases where the grammar is off. For example, '4.1 Evaluation of the storm process improvements', '. . . and the forecasts had negative errors in the accumulated areal rainfall (negative bias)', etc. Please check the whole paper carefully and improve the English language.

10) The plot frame of the Figure 3(b) and Figure 3(c) is not clear.

―――――――――――――――

---

## Referee Comment (RC2) · Y. Xuan (Referee) · 9 Feb 2018

This is a very interesting study which gives a relatively detailed account of how rainfall prediction can be improved using various combination of data assimilation in a widely used model WRF. Some of the findings are certainly of great practical use that may help practitioners to choose proper approach in dealing with severe storms in the context of hydrological forecasting. I also see that the paper is well organised and is easy to read in general.

[Figure]

I do however think more in-depth discussions/findings are needed for the paper to differentiate itself from yet another case study paper. I wish to see that author address my following observations

1. The novelty. I understand the paper is very much a case study on an extreme event which certainly very useful in its own right. I do however think that paper like this should offer certain in-depth findings that will help model development community. I feel that the paper limits itself to present what has come out of the analysis without giving further reasoning on why. For example, the way of using the GTS data is vague and I don't think the author/or the reader have been able to answer why GTS has contributed to the improvement. For example, the location of the observations that have been assimilated, and what kind of variables are used etc. This will help explain the result with deeper understanding.

2. Technical details. There are many combinations in WRF settings that can affect rainfall prediction. A new scheme would have changed the overall conclusion. It would be helpful to discuss this in more details as to why certain schemes are chosen and whether that would affect the final conclusions. Being set as a limited area model, WRF is prone to the impact from the boundary condition. NCEP might be a good and reliable choice, but again, would using data from other centres like CMA and/or ECMWRF change your final conclusion? Further, please make it clear whether the NCEP data has also involved assimilating GTS data in its operational cycle – i.e., whether it an analysis or a forecast initialised at 00hUTC on the day?

3. The hydrological context. Data assimilation is routinely done at various levels in numerical weather prediction. The big problem to produce a hydrologically compatible rainfall forecast is that many of those forecasts fail to capture the two essential aspects: amount and distribution. With reference to the paper, Fig 5 shows a consistent time shift of all the runs in all modes, i.e., the predicted storms started and stopped around 6-h earlier than the actual one. This might be linked to the setting of assimilation, and I suspect that more likely than not it is due to the constraint imposed by the background field from the lateral boundary conditions. This however, has not been properly explored.

4. The choice of using cumulative (only) rainfall may be OK to compare the overall amount in general. Again, for hydrological use, we'd like to see how the prediction agrees with the distributions (both temporal and spatial) of the actual rainfall. So, I think it would be interesting to have a normal hyetograph and a spatial distribution would be more helpful. Some derivative indices like RMSE would make the discussion more convincing.

5. A few terminology and grammar issues: 1) we don't quite often use 'curve' in general, hyetograph is a better and more accurate choice when being used to describe the temporal distribution of rainfall. 2) P5 L26-28 'If more than . . . average value'. This sentence is confusing. 3) P11 L13-15 'The assimilation of radar velocity . . .' I think you meant 'radar radial velocity'. Also the sentence itself is self-contradicting: moisture transport does affect the rainfall 'physical' process. Please elaborate more. 4) P11 L18 '. . . are quite variably' should be 'are quite variable'

---

## Author Comment (AC1) · 9 Feb 2018

Point 1: I agree with the authors that hydrologists are particularly concerned about the accuracy of the accumulative amount and the process of the predicted rainfall at the catchment scale. However, I did not observe any special configuration of WRF or data assimilation for this goal.

Reply: As the reviewer mentioned, the configuration of WRF or data assimilation may have effects on the rainfall prediction, not only the accumulative amount but also the

rainfall process. Before we investigated the Doppler radar and GTS data assimilation, the configuration of WRF has been discussed in detail in our two other articles (Tian et al, 2017a and 2017b), especially for the selection of the WRF physical parameterizations in the same study area of this manuscript. The aim of this study is to explore the potential effects of assimilating different sources of observations from the Doppler weather radar and the Global Telecommunication System (GTS) in improving the mesoscale NWP rainfall products. That is why the eleven modes are specially set for data assimilation. The following sentences are added to address this issue and two references are also added:

"According to our previous investigations on the performances of the most important WRF physical parameterizations affecting the rainfall processes in Northern China (Tian et al, 2017a and 2017b), the most appropriate set of parameterizations for this extreme summer storm , including Kain-Fritsch (KF), WRF single-moment 6 (WSM6) and Mellor-Yamada-Janjic (MYJ), was adopted in this study when configuring the WRF model."

Tian, J., Liu, J., Wang, J., Li, C., Yu, F., and Chu, Z.: A spatio-temporal evaluation of the WRF physical parameterisations for numerical rainfall simulation in semi-humid and semi-arid catchments of Northern China, Atmos. Res., 191: 141-155, doi: 10.1016/j.atmosres.2017.03.012, 2017a.

Tian, J., Liu, J., Yan, D., Li, C., and Yu, F.: Numerical rainfall simulation with different spatial and temporal evenness by using a WRF multiphysics ensemble. Nat. Hazards Earth Syst. Sci., 17: 563-579, doi: 10.5194/nhess-17-563-2017, 2017b.

Point 2: What factors drove your decision to have 40 vertical levels in WRF? Why not more?

Reply: In general, the vertical levels between 25 and 55 are acceptable for numerical weather prediction with the WRF model. The reviewer may think that the vertical levels can affect the performance of WRF model. Actually, the optimal number of the

vertical levels has been deeply investigated by the meteorological society but no consistent conclusion is yet obtained. Aligo et al. (2009) found that the QPF forecasts cannot always be improved by adding the vertical levels with 4-km horizontal resolution in American Midwest. Done et al. (2004) forecasted the convective rainfall in North America with 4-km grid size and the vertical levels were only set at 35. Fierro et al. (2013) simulated a storm event in Oklahoma City, and the horizontal and vertical resolutions were set to be 3-km and 43 levels respectively. Qie et al. (2014) simulated the storm event occurred in Beijing, which is near the study area of this manuscript. The inner domain is 2-km and the vertical levels were set to be 27. Many studies had the horizontal resolution of the WRF inner domain around 3-km as the manuscript, while the vertical levels were less than 40 or close to 40. It is an interesting issue to investigate the relation between the number of the vertical layers and the horizontal resolutions of the WRF model. However, this is not the main concern of this study. We hope to obtain meaningful conclusions with adequate experiments in further studies. The aforementioned references are added in the manuscript to support the use of the 40 layers in this study.

"The two domains were comprised of 40 vertical pressure levels, with the top level set to 50 hPa (Done et al., 2004; Aligo et al, 2009; Fierro et al., 2013; Qie et al, 2014)."

Aligo, E.A., Gallus, W.A., and Segal, M.: On the impact of WRF model vertical grid resolution on Midwest summer rainfall forecasts, Weather Forecast., 24: 575-594, doi: 10.1175/2008WAF2007101.1, 2009.

Qie, X., Zhu, R., Yuan, T., Wu, X., Li, W., and Liu, D.: Application of total-lightning data assimilation in a mesoscale convective system based on the WRF model, Atmos. Res., 145–146: 255-266, doi: 10.1016/j.atmosres.2014.04.012, 2014.

Done, J., Davis, C. A., and Weisman, M.: The next generation of NWP: explicit forecasts of convection using the weather research and forecasting (WRF) model, Atmos. Sci. Lett., 5:110–117, doi: 10.1002/asl.72, 2004.

Fierro, A. O., Mansell, E. R., Macgorman, D. R., and Ziegler, C. L.: The implementation of an explicit charging and discharge lightning scheme within the WRF-ARW model: benchmark simulations of a continental squall line, a tropical cyclone, and a winter storm, Mon. Weather Rev., 141: 2390-2415, doi: 10.1175/MWR-D-12-00278.1, 2013.

Point 3: The uncertainty of weather radar rainfall and GTS data, and their possible influences on the data assimilation process should be carefully specified.

Reply: The main uncertainty of the weather radar rainfall and GTS data comes from the observational and instrumental errors in the data. This can be eliminated through the data quality control procedure before the data assimilated (Rubel and Brugger, 2009). Sokol and Zacharov (2012) also indicated that data with large errors may fail to be assimilated by the NWP model and even make the model crash. In this study, before the data are assimilated by the WRF model, the radar and GTS data were thoroughly checked and strict quality controls were carried out. Descriptions of the data quality control can be found in Line 8-10 and 18-22, Page 6, respectively for the GTS and the radar data.

"The quality control of the GTS data is implemented in WRF-3DVar by defining the observation error covariance. The default US Air Force (AFWA) OBS error file is used in this study, which defines the instrumental and sensor errors for various air, water and surface observation types as well as satellite retrievals."

"The S-band radar belongs to the newest generation weather radar network of China (CINRAD/SC), the quality control of which is supported by China Integrated Meteorological Information Service System (CIMISS) of China Meteorological Administration. The potential error sources, such as ground clutter, radial interference echo, speckles and other artefacts, were removed through the quality control procedure (Tong and Xue, 2005). The rainfall observations from rain gauges and weather radar were also compared to check the quality of the radar data."

Rubel, F., and Brugger, K.: 3-hourly quantitative precipitation estimation over Central

and Northern Europe from rain gauge and radar data, Atmos. Res., 94: 544-554, doi:10.1016/j.atmosres.2009.05.005, 2009.

Sokol, Z., and Zacharov, P.: Nowcasting of precipitation by an NWP model using assimilation of extrapolated radar reflectivity, Q. J. Roy. Meteor. Soc., 138: 1072–1082, doi: 10.1002/qj.970, 2012.

Point 4: The spin-up period may have influences on the rainfall simulation of the NWP. A 6 hour is used in this study, why not 12 h or other times? In fact, the authors should specify or at least discuss all the sense parameterizations, such as the domain design, downscaling ratios ect.

Reply: As for the spin-up period, 6 hour (Givati et al, 2012), 12 hour (Pan et al, 2017) and 24 hour (Jr et al, 2016) are common choices for the WRF model. In this study, all the three periods were tested before the 6 hour was used. The testing results showed that the spin-up periods had little influence on rainfall prediction of the extreme storm event in this study. In order to improve the calculation efficiency, the 6 hour spin-up period was finally adopted. The following sentences are added in the manuscript:

"According to the tests of different spin-up periods (6 h, 12 h and 24 h), which are commonly used in the WRF model (Givati et al, 2012; Pan et al, 2017; Jr et al, 2016), the spin-up periods have little influence on rainfall prediction of the extreme storm event in this study. Considering the calculation efficiency, the 6 hour spin-up period was chosen."

Givati A, Lynn B, Liu Y, and Rimmer A. Using the WRF model in an operational streamflow forecast system for the Jordan River, J. Appl. Meteorol. Clim., 51: 285-299, doi: 10.1175/JAMC-D-11-082.1, 2012.

Pan, X., Li, X., Cheng, G., and Hong, Y.: Effects of 4D-Var data assimilation using remote sensing precipitation products in a WRF model over the complex terrain of an arid region river basin, Remote Sensing, 9: 963, doi: 10.3390/rs9090963, 2017.

Jr, J. H. R., and Johnson, R. H.: On the cumulus diurnal cycle over the tropical warm pool, J. Adv. Model Earth Sy., 8: 669-690, doi: 10.1002/2015MS000610, 2016.

Besides the spin-up period, the performance of the WRF model can also be influenced by the domain design and the downscaling ratios. In this study, the outer model domain covers Eastern China, Bohai Sea, Yellow Sea and parts of East China Sea, and the inner domain covers Beijing-Tianjin-Hebei region. The large scale topography and the main climate zone can be covered by the nested domains. That is to say, the main influence factors of the rainfall formation are considered in this study through the domain design. The inner domain is set as 3-km, which references to the study from Wang et al. (2012). Both studies focus on the Beijing-Tianjin-Hebei region and assimilate the radar data to improve the rainfall prediction. The downscaling ratio of 1:3 is fixed in the previous generation of the WRF model, MM5. With WRF the downscaling ratio can be flexible. Although some studies indicate that the commonly used downscaling ratio of 1:3 do not always perform the best for all kinds of rainfall events, it is still the best choice as a whole (Liu et al, 2012). The following sentences and two references are added in Line 21-22 and Line 23-24, Page 4:

". . . the downscaling ratio was set to 1:3, which was commonly used and always performed well (Liu et al, 2012; Yang et al, 2012; Chambon et al, 2014)."

"The large scale topography and the main climate zone can be covered by the nested domains (Wang et al, 2012)."

Liu, J., Bray, M., and Han, D.: Sensitivity of the Weather Research and Forecasting (WRF) model to downscaling ratios and storm types in rainfall simulation, Hydrol. Process., 26, 3012-3031, doi: 10.1002/hyp.8247, 2012.

Wang, H., Sun, J., Fan, S., and Huang, X.: Indirect assimilation of radar reflectivity with WRF 3D-Var and its impact on prediction of four summertime convective events. J. Appl. Meteorol. Clim., 52: 889-902, doi: 10.1175/JAMC-D-12-0120.1, 2013.

Point 5: It is necessary to explain the causes of the heavy rainfall event in this study. The readers may be interested in the very convective system which affects the choices of the WRF parameterisation schemes.

Reply: Thanks for the reviewer's suggestion. The following paragraph is added to supplement the background information of the storm event in the manuscript:

"Many studies have investigated the causes and the properties of the storm (Sang et al, 2013; Zhong et al, 2015). Before the extreme storm event took place, an encounter between a northward-moving subtropical high vortex and an eastward-moving cold vortex in the mid-high troposphere provided a stable atmospheric circulation over the study area, which was conducive to heavy rain formation. Abundant water vapor transported from a low-level jet, strong upward motion caused by Taihang Mountain, together with a long duration of dense air humidity were the primary causes of the storm event. The extreme storm event contained two phases: 1) a strong convective rain that occurred in the warm sector ahead of the cold front and 2) a dominant frontal rain after the arrival of the cold front (Guo et al, 2015). Those showed the reasons why the parameterizations of KF, WSM6 and MYJ were chosen for WRF rainfall prediction. KF has strong ability in simulating the low-level jet and the upward transportation of vapour (Kain, 2003). WSM6 contains six water substance variables, which can realistically identify rainfall formation (Kim et al, 2013). MYJ is more suitable for the simulation of the convection system (Janjić, 1994)."

Sang, Y.F., Wang, Z., and Liu, C.: What factors are responsible for the Beijing storm, Nat. Hazards., 65: 2399-2400, doi: 10.1007/s11069-012-0426-8, 2013.

Zhong, L., Mu, R., Zhang, D., Zhao, P., Zhang, Z., and Wang, N.: An observational analysis of warm-sector rainfall characteristics associated with the 21 July 2012 Beijing extreme rainfall event, J. Geophys. Res., 120: 3274-3291. doi: 10.1002/2014JD022686, 2015.

Guo, C., Xiao, H., Yang, H., and Tang, Q.: Observation and modeling analyses of the

macro-and microphysical characteristics of a heavy rain storm in Beijing, Atmos. Res., 156: 125-141, doi: 10.1016/j.atmosres.2015.01.007, 2015.

Kain, J.S.: The Kain-Fritsch convective parameterization: an update, J. Appl. Meteorol., 43: 170-181, doi: 10.1175/1520-0450(2004)043<0170:TKCPAU>2.0.CO;2, 2004.

Kim, J.H., Shin, D.B., and Kummerow, C.: Impacts of a priori databases using six WRF microphysics schemes on passive microwave rainfall retrievals, J. Atmos. Ocean. Technol., 30: 2367–2381, doi: 10.1175/JTECH-D-12-00261.1, 2013.

Janjić, Z.I.: The step-mountain eta coordinate model: further developments of the convection, viscous sublayer, and turbulence closure schemes, Mon. Weather Rev., 122: 927–945, doi: 10.1175/1520-0493(1994)122<0927:TSMECM>2.0.CO;2, 1994.

Point 6: The description of assimilation mechanism for the radar data is too simple. Please complement the model-derived observation operators that are adopted in the 3DVar method. This may explain why the assimilation of the radar reflectivity performs better than the radial velocity.

Reply: The reviewer's point is very important and radar observation operators are added in the manuscript:

"2.3 Radar observation operators: radar reflectivity and radial velocity

The total water mixing ratio qt was chosen as the moisture control variable instead of the pseudo-relative humidity when assimilating the radar reflectivity and radial velocity. Dudhia (1989) provided a warm parameterization, which assists the partitioning of the rainwater mixing ratio qr, the cloud water mixing ratio qc, and the water vapor mixing ratio qv. Eq. (2) shows the observation operator used to calculate the model-derived radar reflectivity Z from the rainwater mixing ratio qr (Sun and Crook, 1997):

Equation (2)

where  is the density of air. By assuming a Marshall-Palmer raindrop size distribution

and that the ice phases have no effect on reflectivity, Eq. (2) can be derived.

For the assimilation of the radial velocity, the preconditioned wind control variables were also combined with the rainwater mixing ratio qr. Eq. (3)-(5) show how the model-derived radial velocity Vr was calculated:

Equation (3)

where u, v and w represent the three-dimensional wind field, x, y, and z represent the location of the observation point, and xi, yi, and zi represent the location of the radar station. ri is the distance between the data point and the radar, and vt is the hydrometer fall speed or terminal velocity.

According to Sun and Crook (1998), vt can be represented as follows:

Equation (4)

Equation (5)

where a is the correction factor, p is the base-state pressure, and p0 is the pressure at the ground."

Dudhia, J.: Numerical study of convection observed during the winter monsoon experiment using a mesoscale two-dimensional model, J. Atmos. Sci., 46: 3077-3107, doi: 10.1175/1520-0469(1989)046<3077:NSOCOD>2.0.CO;2, 1989.

Sun, J., and Crook, N. A.: Dynamical and microphysical retrieval from Doppler radar observations using a cloud model and its adjoint. Part I: Model development and simulated data experiments, J. Atmos. Sci., 54: 1642-1661. doi: 10.1175/1520-0469(1997)054<1642:DAMRFD>2.0.CO;2, 1997.

Sun, J., and Crook, N. A.: Dynamical and microphysical retrieval from Doppler radar observations using a cloud model and its adjoint. Part II: Retrieval experiments of an observed Florida convective storm, J. Atmos. Sci., 55: 835-852, doi: 10.1175/1520-0469(1998)055<0835:DAMRFD>2.0.CO;2, 1998.

Point 7: In general, this study describes the results of different data assimilation experiments and explains the reasons why the assimilation of the radar velocity always leads to worse results. It is wondered why the GTS data and the radar reflectivity can perform better, and neither does the author give a plausible explanation why the assimilation of GTS data together with the radar data performs the best among the eleven assimilation modes. The revised manuscript needs to contain more deep analysis in the Discussion section.

Reply: The suggestion is very helpful for improving the manuscript and the readers may also have the same query while reading the paper. The following sentences are added in the manuscript:

"Assimilating radar reflectivity always had a positive effect on the forecasted rainfall, and the performance was relatively stable. The data assimilation modes which involved the radar reflectivity always performed better than the others. The main reason is that radar reflectivity data contain information related to the precipitation hydrometeors. According to the Eq. (2), the assimilation of radar reflectivity is a correction to the humidity field in essence, which directly influences the rainfall prediction. Additionally, Figure 3 shows that the radar reflectivity used in this study had good quality, which helped result in more effective assimilation results."

"Although the spatial density of the GTS data is relatively low compared to the radar data, it contains observations of various atmospheric states, including temperature, humidity, pressure, wind speed, etc. Assimilating the GTS data can update the atmospheric background holistically. With the two-way nesting mechanism, data assimilation in the outer domain can help improve the atmospheric motion and water vapour transport in the inner domain. When the GTS data and the radar data are assimilated at the same time, the atmospheric motion in the large scale can be improved by the assimilation of GTS data, while the humidity field in the smaller scale can be corrected by the assimilation of radar data. That is why the assimilation of GTS data together with the radar data performs the best in this study."

Point 8: The ultimate goal of the WRF applications in this study is for flood forecasts. It would be helpful to add references to explain how much the flood forecasts in general can be improved by the improvement of the rainfall accuracy. I also look forward to the follow-up study for the improvement of flood forecasts with data assimilation.

Reply: Thanks for the reviewer's suggestion. The following paragraph is added in the manuscript:

"Ultimately, the main goal of rainfall prediction based on the WRF model is to make the flow forecasts. Errors in the forecasted rainfall process and amount can result in divergent flood peak time and peak stage of the flood (Shih et al, 2014). Therefore, data assimilation is an important tool in improving the forecasted rainfall as well as the flow. Yucel et al. (2015) assimilated conventional meteorological observations to improve the rainfall prediction, and the mean runoff error was reduced by 14.7% with data assimilation in the Black Sea Region. While in the work of Rossa et al. (2010), the error of simulated peak discharge can be reduced from 50% to 14% when the radar data were assimilated in the hydro-meteorological model in the Dese river catchment. In further study, the coupled atmospheric-hydrological model with data assimilation will be built and flood forecasts from the coupling system will be examined to evaluate the effect of data assimilation on flood forecasting."

Shih, D. S., Chen, C. H., Yeh, G. T.: Improving our understanding of flood forecasting using earlier hydro-meteorological intelligence, J. Hydrol., 512: 470-481, doi: 10.1016/j.jhydrol.2014.02.059, 2014.

Yucel, I., Onen, A., Yilmaz, K. K., Gochis, D. J.: Calibration and evaluation of a flood forecasting system: Utility of numerical weather prediction model, data assimilation and satellite-based rainfall, J. Hydrol., 523: 49-66, doi: 10.1016/j.jhydrol.2015.01.042, 2015.

Rossa, A. M., Guerra, F. L. D., Borga, M., Zanon, F., Settin, T., Leuenberger, D.: Radar-driven high-resolution hydro-meteorological forecasts of the 26 September 2007

Venice flash flood, J. Hydrol., 394: 230-244, doi: 10.1016/j.jhydrol.2010.08.035, 2010.

Point 9: There are several typos and some cases where the grammar is off. For example, '4.1 Evaluation of the storm process improvements', '... and the forecasts had negative errors in the accumulated areal rainfall (negative bias)', etc. Please check the whole paper carefully and improve the English language.

Reply: Grammar and spelling errors are corrected in the revised manuscript. We will make efforts to further improve the readability of the paper.

Point 10: The plot frame of the Figure 3(b) and Figure 3(c) is not clear.

Reply: The figures are revised as followed.
* * *
[Figure]

[Figure]

Figure 3. Comparison of the rainfalls from the rain gauges and radar: (a) time series bars of the hourly catchment areal rainfall; (b) 24 h rainfall accumulation from the rain gauges; (c) 24 h rainfall a 5 ccumulation from the radar.

**Fig. 1.**

$$Z = 43.1 + 17.5\log(\rho q_r) \tag{2}$$

$$V_r = u\,\frac{x - x_i}{r_i} + v\,\frac{y - y_i}{r_i} + \left(w - v_i\right)\frac{z - z_i}{r_i} \tag{3}$$

$$v_i = 5.40a(\rho q_r)^{0.125} \tag{4}$$

$$a = \left(\frac{p_0}{\bar{p}}\right)^{0.4} \tag{5}$$

**Fig. 2.**

---

## Author Comment (AC2) · 25 Feb 2018

We appreciate the referee's high evaluation of the paper. We hope it can inspire more and wider studies seeking for the efficient way of data assimilation by making use of various sources of observations. We also hope it can promote involving easy data assimilation in the hydrological applications of the numerical weather prediction models. We would like to further improve the rigorousness and the depth of the paper according to the referee's helpful suggestions. Below are the point-to-point replies to the comments.

[Figure]

Comments:

Point 1: I understand the paper is very much a case study on an extreme event which certainly very useful in its own right. I do however think that paper like this should offer certain in-depth findings that will help model development community. I feel that the paper limits itself to present what has come out of the analysis without giving further reasoning on why. For example, the way of using the GTS data is vague and I don't think the author/or the reader have been able to answer why GTS has contributed to the improvement. For example, the location of the observations that have been assimilated, and what kind of variables are used etc. This will help explain the result with deeper understanding.

Reply: Thanks for the reviewer's suggestion, which is very helpful in explaining the contribution of the GTS data to the rainfall improvement. The following paragraph will be added in the manuscript and the Figure 2 is updated by adding the locations of the GTS data:

"In this study, five GTS datasets, including SOUND, SYNOP, PILOT, AIREP and METAR, were assimilated in the WRF model. Detailed descriptions of the datasets are shown in a new table. According to Figure 2, the observations covered by the outer domain were mostly located on land and only a few were on the ocean. The data located on land were distributed evenly, which is very helpful for the stability of the WRF model during data assimilation (Carrassi et al, 2008). The SOUND and SYNOP data took the majority of the GTS data, which means that the observations from surface-based observing station and upper-air observatory have the most contribution to the improvement of rainfall prediction. Pressure, temperature, humidity and wind from the surface and upper air are contained in SOUND and SYNOP datasets. The assimilation of these meteorological elements can directly correct the initial and lateral boundary conditions through the wide horizontal coverage and high vertical levels (Tu et al, 2017)."

References:

Carrassi, A., Ghil, M., Trevisan, A. and Uboldi, F.: Data assimilation as a nonlinear dynamical systems problem: stability and convergence of the prediction-assimilation system. Chaos, 18: 023112, doi: 10.1063/1.2909862, 2008.

Tu, C.C., Chen, Y.L., Chen, S.Y., Kuo, Y.H., and Lin, P.L.: Impacts of Including Rain-Evaporative Cooling in the Initial Conditions on the Prediction of a Coastal Heavy Rainfall Event during TiMREX, Mon. Weather Rev., 145: 253-277, doi: doi:10.1175/MWR-D-16-0224.1, 2017.

Point 2: There are many combinations in WRF settings that can affect rainfall prediction. A new scheme would have changed the overall conclusion. It would be helpful to discuss this in more details as to why certain schemes are chosen and whether that would affect the final conclusions. Being set as a limited area model, WRF is prone to the impact from the boundary condition. NCEP might be a good and reliable choice, but again, would using data from other centres like CMA and/or ECMWF change your final conclusion? Further, please make it clear whether the NCEP data has also involved assimilating GTS data in its operational cycle – i.e., whether it an analysis or a forecast initialised at 00hUTC on the day?

Reply: As the reviewer mentioned, the combinations in WRF settings can affect the rainfall prediction. Before we investigated the assimilating of Doppler radar and GTS data, the WRF settings have been discussed in detail in our two other articles (Tian et al, 2017a and 2017b), especially for the selection of the WRF physical parameterizations in the same study area of this manuscript. The WRF model settings are adjusted to the best for the rainfall prediction. This study is aimed to explore the potential effects of assimilating different sources of observations from the Doppler weather radar and the Global Telecommunication System (GTS) in improving the mesoscale NWP rainfall products. The following sentences will be added in the manuscript:

"According to our previous investigations on the performances of the most important WRF physical parameterizations affecting the rainfall processes in Northern China

(Tian et al, 2017a and 2017b), the most appropriate set of parameterizations for this extreme summer storm , including Kain-Fritsch (KF), WRF single-moment 6 (WSM6) and Mellor-Yamada-Janjic (MYJ), was adopted in this study when configuring the WRF model."

References:

Tian, J., Liu, J., Wang, J., Li, C., Yu, F., and Chu, Z.: A spatio-temporal evaluation of the WRF physical parameterisations for numerical rainfall simulation in semi-humid and semi-arid catchments of Northern China, Atmos. Res., 191: 141-155, doi: 10.1016/j.atmosres.2017.03.012, 2017a.

Tian, J., Liu, J., Yan, D., Li, C., and Yu, F.: Numerical rainfall simulation with different spatial and temporal evenness by using a WRF multiphysics ensemble. Nat. Hazards Earth Syst. Sci., 17: 563-579, doi: 10.5194/nhess-17-563-2017, 2017b.

The initial and lateral boundary conditions provided by different centres like NCEP, CMA and ECMWF may make some difference of the rainfall forecasts. Some studies have specialized the different performance of the WRF model based on the initial and lateral boundary conditions from the different centres (Srivastava et al, 2013; Islam et al, 2015). Before the NCEP data was used in this study, we also tests ECMWF for data assimilation with storm events in the same region. Although the rainfall forecasts showed a little different based on the boundary conditions from the two centres, the patterns of improvements from different data assimilation modes are quite similar and the same conclusions can be obtained. Some study also found that the boundary conditions from different centres could even lead to similar rainfall forecasts through an optimal control (Zou, 1996). In order to highlight the main purpose of this study, we only present the assimilation results using the NCEP data. We appreciate the referee's deep insights and we also hope our work can inspire further studies on testing the data assimilation effects using other boundary data, such as CMA. The following sentences will be added in the Discussion section of the manuscript:

"Before the NCEP driven data was used in this study, ECMWF was also tested for the data assimilation with the same storm event. Although the rainfall forecasts showed some differences based on the boundary conditions from the two centres, the improvement patterns from different data assimilation modes were quite similar. The initial and lateral boundary conditions do have some potential impact on the rainfall forecasts results. More studies should be carried out to verify the effects of data assimilations using different driven data."

References:

Zou, X.: Rainfall assimilation through an optimal control of initial and boundary conditions in a limited-area mesoscale model, Mon. Weather Rev., 124: 2859-2882, doi: 10.1175/1520-0493(1996)124<2859:RATAOC>2.0.CO;2, 1996.

Srivastava, P.K., Han, D., Ramirez, M.A.R., and Islam, T.: Comparative assessment of evapotranspiration derived from NCEP and ECMWF global datasets through Weather Research and Forecasting model, Atmos. Sci. Lett., 14: 118-125, doi: 10.1002/asl2.427, 2013.

Islam, T., Srivastava, P.K., Rico-Ramirez, M.A., Dai, Q., Gupta, M., and Singh, S.K.: Tracking a tropical cyclone through WRF-ARW simulation and sensitivity of model physics, Nat. Hazards, 76: 1473-1495, doi: 10.1007/s11069-014-1494-8, 2015.

The NCEP data (GFS) is the forecast data which is initialised at 00hUTC on the day and the GTS data are not assimilated in GFS, which can also be proven by the improvement of the rainfall forecasts with the assimilation of the GTS data in the outer domain. This will be further clarified in the revised manuscript.

Point 3: Data assimilation is routinely done at various levels in numerical weather prediction. The big problem to produce a hydrologically compatible rainfall forecast is that many of those forecasts fail to capture the two essential aspects: amount and distribution. With reference to the paper, Fig 5 shows a consistent time shift of all the runs

in all modes, i.e., the predicted storms started and stopped around 6-h earlier than the actual one. This might be linked to the setting of assimilation, and I suspect that more likely than not it is due to the constraint imposed by the background field from the lateral boundary conditions. This however, has not been properly explored.

Reply: We agree with the referee's opinion. The time shift of the storm was more likely caused by the background field from the lateral boundary conditions. From Fig 5 it can be seen that the 6h shift starts with run1, which is the original WRF run without data assimilation. Some studies also indicate that when data containing the information of water vapor are assimilated in the numerical weather prediction model, the predicted rainfall may start and stop earlier than the actual one (Georgakakos, 2000; Sun et al, 2016). The main reason is that the information of water vapor can make the rain in the initial fields form and fall to the earth more quickly in the case of no matched dynamic fields (Sun, 2005). In this study, both the radar reflectivity and the GTS data contain the information of water vapor. In addition, the radial velocity can neither improve the dynamic field through data assimilation. All the above reasons result in a consistent 6h time shift of all the runs with or without data assimilation in Fig 5. Considering the case, an error prediction model could be built to correct the consistent error. Some studies also suggest the assimilation of the latent heat might help improve the start and ending time of the forecasted rainfall process (Stephan et al, 2010; Schraff et al, 2016). The time shift issue will be addressed by adding the following paragraph in the Discussion section:

"It can be found in Figure 5 that the predicted storms always start and end around 6-h earlier than the observations. Besides the errors in the boundary conditions, it is found that the assimilation of the water vapor information (contained in the radar reflectivity and the GTS data) can make the rain in the initial fields form and fall to the earth more quickly (Georgakakos, 2000; Sun, 2005; Sun et al, 2016). Considering the error is consistent, an error prediction model could be built in further studies, and the assimilation of the latent heat may also be helpful in correcting the starting and ending

time of the forecasted rainfall process (Stephan et al, 2010; Schraff et al, 2016)."

References:

Georgakakos, K.P.: Covariance propagation and updating in the context of real-time radar data assimilation by quantitative precipitation forecast models, J. Hydrol., 239: 115-129, doi: 10.1016/S0022-1694(00)00355-3, 2000.

Sun, J., Wang, H., Tong, W., Zhang, Y., Lin, C.Y., and Xu, D.: Comparison of the impacts of momentum control variables on high-resolution variational data assimilation and precipitation forecasting, Mon. Weather Rev., 144: 149-169, doi: 10.1175/MWR-D-14-00205.1, 2016.

Sun, J.: Convective-scale assimilation of radar data: progress and challenges, Q. J. Roy. Meteor. Soc., 131: 3439–3463, doi: 10.1256/qj.05.149, 2005.

Stephan, K., Klink, S., and Schraff, C.: Assimilation of radar-derived rain rates into the convective-scale model COSMO-DE at DWD, Q. J. Roy. Meteor. Soc., 134: 1315-1326, doi: 10.1002/qj.269, 2010.

Schraff, C., Reich, H., Rhodin, A., Schomburg, A., Stephan, K., Perianez, A. and Potthast, R. Kilometre-scale ensemble data assimilation for the COSMO model (KENDA), Q. J. Roy. Meteor. Soc., 142: 1453-1472, doi: 10.1002/qj.2748, 2016.

Point 4: The choice of using cumulative (only) rainfall may be OK to compare the overall amount in general. Again, for hydrological use, we'd like to see how the prediction agrees with the distributions (both temporal and spatial) of the actual rainfall. So, I think it would be interesting to have a normal hyetograph and a spatial distribution would be more helpful. Some derivative indices like RMSE would make the discussion more convincing.

Reply: The referee's suggestion is thoughtful. Both the temporal and spatial distributions of the rainfall can have potential impacts on the formation of the flow, thus are paid equally important attention by the hydrologists. Actually when we initially organized the

paper, we intended to present the spatial distributions as well as the temporal variations of the forecasted rainfall from different data assimilation modes. However, that would involve too many figures (considering there are 11 assimilation modes), and we have also noticed that the lumped rainfall-runoff models are still widely used by the hydrological community and proven to be even better than the distributed ones in producing the forecasted flow. So in this paper we only paid special attention to the cumulative process of the areal rainfall across the catchment. However, if the number of figures is not limited by the journal, we would like to add the normal hyetographs and the spatial distributions of the forecasted rainfall, at least those from some representative assimilation modes.

Point 5: A few terminology and grammar issues: 1) we don't quite often use 'curve' in general, hyetograph is a better and more accurate choice when being used to describe the temporal distribution of rainfall. 2) P5 L26-28 'If more than . . . average value'. This sentence is confusing. 3) P11 L13-15 'The assimilation of radar velocity . . .' I think you meant 'radar radial velocity'. Also the sentence itself is self-contradicting: moisture transport does affect the rainfall 'physical' process. Please elaborate more. 4) P11 L18 '. . . are quite variably' should be 'are quite variable'.

Reply: The terminology and grammar errors will be carefully checked and corrected in the revised manuscript. We will also make efforts to further improve the readability of the paper. The four issues mentioned above are addressed as follows:

1) The terminology 'curve' will be replaced by 'hyetograph' throughout the manuscript.

2) The sentence is revised as: "The forecasted areal rainfall is calculated by averaging values of the grid cells those have more than 50% area located inside the Zijingguan catchment."

3) The sentence is revised as: "The assimilation of radar radial velocity cannot directly influence the physical process of rainfall formation, although the assimilation can change the wind field and affect the water vapor transport."

4) Revised accordingly.

The descriptions of the five datasets

| Name | Meaning of the dataset |
|---|---|
| SOUND | Upper-level pressure, temperature, humidity and wind report |
| SYNOP | Report of surface observation from a fixed land station |
| PILOT | Upper-wind report from a fixed land station |
| AIREP | Aircraft weather report |
| METAR | Aerodrome routine meteorological report |

**Fig. 1.**

[Figure]

**Figure 2.** Locations of the radar scan area, the GTS data, the study catchments and the two nested domains.

**Fig. 2.**

---

## Author Comment (AC3) · 26 Feb 2018

In the above reply to Referee #2, the description table of the assimilated GTS data is corrected as follows.

[Figure]

| Dataset | Descriptions |
|---------|--------------|
| SOUND | Upper-level pressure, temperature, humidity and wind report from a fixed or mobile land station, a sea station or a sonde released by carrier balloons or aircraft. |
| SYNOP | Report of surface observation from a fixed or mobile land station. |
| PILOT | Upper-wind report from a fixed or mobile land station or a sea station. |
| AIREP | Aircraft weather report. |
| METAR | Aerodrome routine or special meteorological report. |

**Fig. 1.** Descriptions of the GTS datasets assimilated in the study.

---

## Author Response (AR1)

Dear editor:

Thank you very much for giving us the opportunity to revise the manuscript. We apologise for the extension of the revision process, due to our insistence on trying the best to make the work satisfactory. Now it is believed that all necessary changes are made to address every point of the referees' concerns. Major attentions are paid to solve two key issues. Firstly, in-depth analyses are provided with a better reasoning for the different model performance when assimilating different observations. We also tried to find the reason why the best improvement is found when the GTS data and the radar reflectivity is assimilated at the same time. Please see the reply to Point 1 of Referee 2 or the reply to Point 7 of Referee 1 for more details. The discussion section is rearranged with Table 1, Figure 2 and Figure 8 added or revised. Secondly, a new section (Section 4.3) is added, where both the spatial and the temporal distributions of the forecasted rainfall are presented and evaluated quantitatively. Two new figures and one new table (Figure 6, Figure 7 and Table 4) are added in the section. Please refer to the reply to Point 4 of Referee 2 for details. Efforts are also made to improve the readability of the paper. Spelling errors are corrected and the references are updated. We hope the manuscript can be found rigorously and sufficiently improved.

During the revision, we are encouraged by the positive comments from Referee #1, "*the results of this paper are helpful for improving the rainfall prediction from the hydrological perspective and the content is of interest to the readers of the HESS journal*", and those from Referee #2 "*this is a very interesting study which gives a relatively detailed account of how rainfall prediction can be improved using various combination of data assimilation in a widely used model WRF ... some of the findings are certainly of great practical use that may help practitioners to choose proper approach in dealing with severe storms in the context of hydrological forecasting*".

Our study aims at seeking for the effective way of data assimilation by making use of the weather radar together with traditional meteorological observations. We hope our study can draw more attentions from the hydrological community to the appropriate utilisation of data assimilation in generating accurate and reliable NWP rainfall predictions for hydrological use. We would like to further improve the rigorousness and the depth of the paper according to any additional requirements from you and the referees .

We are looking forward to your response.
Yours sincerely,

Dr. Jia Liu

Key State Laboratory of Simulation and Regulation of Water Cycle in River Basin
China Institute of Water Resources and Hydropower Research
No.1 Yu-Yuan-Tan South Street, Haidian district, Beijing 100038
E-mail: hettyliu@126.com, jia.liu@iwhr.com

**Evaluation of Doppler radar and GTS data Assimilation for NWP Rainfall Prediction of an Extreme Summer Storm in Northern China: from the Hydrological Perspective**

**Reply to Referee #1**

**Comments:**

**Point 1:** I agree with the authors that hydrologists are particularly concerned about the accuracy of the accumulative amount and the process of the predicted rainfall at the catchment scale. However, I did not observe any special configuration of WRF or data assimilation for this goal.

**Reply:** As the reviewer mentioned, the configuration of WRF or data assimilation may have effects on the rainfall prediction, not only the accumulative amount but also the rainfall process. Before we investigated the Doppler radar and GTS data assimilation, the configuration of WRF has been discussed in detail in our two other articles (Tian et al, 2017a and 2017b), especially for the selection of the WRF physical parameterizations in the same study area of this manuscript. The aim of this study is to explore the potential effects of assimilating different sources of observations in improving the mesoscale NWP rainfall products. That is why the 11 modes are designed for data assimilation. The following sentences are added to address this issue and two references are also added in Line 32, Page 4 and Line 1-4, Page 5:

*"According to our previous investigations on the performances of the most important WRF physical parameterizations affecting the rainfall processes in Northern China (Tian et al, 2017a and 2017c), the most appropriate set of parameterizations for this extreme summer storm , including Kain-Fritsch (KF), WRF single-moment 6 (WSM6) and Mellor-Yamada-Janjic (MYJ), was adopted in this study when configuring the WRF model."*

References:

Tian, J., Liu, J., Wang, J., Li, C., Yu, F., and Chu, Z.: A spatio-temporal evaluation of the WRF physical parameterisations for numerical rainfall simulation in semi-humid and semi-arid catchments of Northern China, Atmos. Res., 191, 141-155, doi: 10.1016/j.atmosres.2017.03.012, 2017a.

Tian, J., Liu, J., Yan, D., Li, C., and Yu, F.: Numerical rainfall simulation with different spatial and temporal evenness by using a WRF multiphysics ensemble. Nat. Hazards Earth Syst. Sci., 17, 563-579, doi: 10.5194/nhess-17-563-2017, 2017b.

The main reason for the selection of the WRF physical parameterizations are also added in Line 26-30, Page 6:

*"Those showed the reasons why the parameterizations of KF, WSM6 and MYJ were chosen for*

*WRF rainfall prediction. KF has strong ability in simulating the low-level jet and the upward transportation of vapour (Kain, 2004). WSM6 contains six water substance variables, which can realistically identify rainfall formation (Kim et al, 2013). MYJ is more suitable for the simulation of the convection system (Janjić, 1994)."*

References:

Kain, J.S.: The Kain-Fritsch convective parameterization: an update, J. Appl. Meteorol., 43, 170-181, doi: 10.1175/1520-0450(2004)043<0170:TKCPAU>2.0.CO;2, 2004.

Kim, J.H., Shin, D.B., and Kummerow, C.: Impacts of a priori databases using six WRF microphysics schemes on passive microwave rainfall retrievals, J. Atmos. Ocean. Technol., 30, 2367–2381, doi: 10.1175/JTECH-D-12-00261.1, 2013.

Janjić, Z.I.: The step-mountain eta coordinate model: further developments of the convection, viscous sublayer, and turbulence closure schemes, Mon. Weather Rev., 122, 927–945, doi: 10.1175/1520-0493(1994)122<0927:TSMECM>2.0.CO;2, 1994.

**Point 2:** What factors drove your decision to have 40 vertical levels in WRF? Why not more?

**Reply:** In general, the vertical levels between 25 and 55 are acceptable for numerical weather prediction with the WRF model. The reviewer may feel that the vertical levels affects the performance of WRF model. Actually, the optimal number of the vertical levels was deeply investigated by the meteorological society but no consistent conclusion has yet been obtained. Aligo et al. (2009) found that the QPF forecasts cannot always be improved by adding the vertical levels with 4-km horizontal resolution in American Midwest. Done et al. (2004) forecasted the convective rainfall in North America with 4-km grid size and the vertical levels were only set at 35. Fierro et al. (2013) simulated a storm event in Oklahoma city, and the horizontal and vertical resolutions were set to be 3-km and 43 levels respectively. Qie et al. (2014) simulated the storm event occurred in Beijing, which was near the study area of this manuscript. The inner domain was 2-km and the vertical levels were set to be 27. Many studies had the horizontal resolution of the WRF inner domain around 3-km the same as the manuscript, while the vertical levels were less than 40 or close to 40. It is an interesting issue to investigate the relation between the number of the vertical layers and the horizontal resolutions of the WRF model. However, this is not the main concern of this study. We hope to obtain meaningful conclusions with adequate experiments in further studies. The aforementioned references are added in Line 26, Page 4 to support the use of the 40 layers in this study.

*"The two domains were comprised of 40 vertical pressure levels, with the top level set to 50 hPa (Done et al., 2004; Aligo et al, 2009; Fierro et al., 2013; Qie et al, 2014)."*

**Point 7:** In general, this study describes the results of different data assimilation experiments and explains the reasons why the assimilation of the radar velocity always leads to worse results. It is

wondered why the GTS data and the radar reflectivity can perform better, and neither does the author give a plausible explanation why the assimilation of GTS data together with the radar data performs the best among the eleven assimilation modes. The revised manuscript needs to contain more deep analysis in the Discussion section.

**Reply:** The suggestion is very helpful for improving the manuscript and the readers may also have the same query while reading the paper. The following paragraphs together with Fig. 8 are added in the Line 21-32, Page 13, and Line 1-18, Page 14:

[revised manuscript text omitted]

**Point 8:** The ultimate goal of the WRF applications in this study is for flood forecasts. It would be helpful to add references to explain how much the flood forecasts in general can be improved by the improvement of the rainfall accuracy. I also look forward to the follow-up study for the improvement of flood forecasts with data assimilation.

**Reply:** The following paragraph is added in Line 34, Page 14 and Line 1-8, Page 15:

*"The ultimate goal for the application of the numerical rainfall prediction is to make flow forecasts at the catchment outlet. Errors in the forecasted rainfall process and the accumulative amount can result in divergent flood peak time and peak stage of the flood (Shih et al, 2014). Therefore, data assimilation is an important tool in improving the forecasted rainfall as well as the flow. Yucel et al. (2015) assimilated conventional meteorological observations to improve the rainfall prediction, meanwhile the mean runoff error was reduced by 14.7% with data assimilation in the Black Sea Region. The peak discharge error was also reduced from 50% to 14% when the radar data were assimilated in a hydro-meteorological model built in the Dese river catchment (Rossa et al., 2010). In the further study, the coupled atmospheric-hydrological model with data assimilation will be built and flood forecasts from the coupling system will be examined to evaluate the effect of data assimilation on flood forecasting."*

**Point 9:** There are several typos and some cases where the grammar is off. For example, '4.1 Evaluation of the storm process improvements', '… and the forecasts had negative errors in the accumulated areal rainfall (negative bias)', etc. Please check the whole paper carefully and improve the English language.

**Reply:** Grammar and spelling errors are corrected in the revised manuscript, and efforts are also made to further improve the readability of the paper.

**Point 10:** The plot frame of the Figure 3(b) and Figure 3(c) is not clear.

**Reply:** The figures are revised as followed.

[Figure]

**Evaluation of Doppler radar and GTS data Assimilation for NWP Rainfall Prediction of an Extreme Summer Storm in Northern China: from the Hydrological Perspective**

**Reply to Referee #2**

**Comments:**

**Point 1:** I understand the paper is very much a case study on an extreme event which certainly very useful in its own right. I do however think that paper like this should offer certain in-depth findings that will help model development community. I feel that the paper limits itself to present what has come out of the analysis without giving further reasoning on why. For example, the way of using the GTS data is vague and I don't think the author/or the reader have been able to answer why GTS has contributed to the improvement. For example, the location of the observations that have been assimilated, and what kind of variables are used etc. This will help explain the result with deeper understanding.

**Reply:** Thanks for the reviewer's suggestion. In the revised manuscript, efforts are made from all possible aspects to explain contribution of the GTS data in the rainfall improvement. The following sentences are added in Line 12-16, Page 7 to introduce the variables of GTS data assimilated in this study and Fig. 2 is updated by adding the locations of the GTS data:

[revised manuscript text omitted]

**Point 2:** There are many combinations in WRF settings that can affect rainfall prediction. A new scheme would have changed the overall conclusion. It would be helpful to discuss this in more details as to why certain schemes are chosen and whether that would affect the final conclusions. Being set as a limited area model, WRF is prone to the impact from the boundary condition. NCEP might be a good and reliable choice, but again, would using data from other centres like CMA and/or ECMWF change your final conclusion? Further, please make it clear whether the NCEP data has also involved assimilating GTS data in its operational cycle – i.e., whether it an analysis or a forecast initialised at 00hUTC on the day?

**Reply:** As the reviewer mentioned, the combinations in WRF settings can affect the rainfall prediction. Before we investigated the assimilating of Doppler radar and GTS data, the WRF settings have been discussed in detail in our two other articles (Tian et al, 2017a and 2017c), especially for the selection of the WRF physical parameterizations in the same study area of this manuscript. The WRF model settings are adjusted to the best for the rainfall prediction. This study is aimed to explore the potential effects of assimilating different sources of observations from the Doppler weather radar and the Global Telecommunication System (GTS) in improving the mesoscale NWP rainfall products. The following sentences are added to address the issue in Line 32, Page 4 and Line 1-4, Page 5:

*"According to our previous investigations on the performances of the most important WRF physical parameterizations affecting the rainfall processes in Northern China (Tian et al, 2017a and 2017c), the most appropriate set of parameterizations for this extreme summer storm , including Kain-Fritsch (KF), WRF single-moment 6 (WSM6) and Mellor-Yamada-Janjic (MYJ), was adopted in this study when configuring the WRF model (Miao et al, 2011; Guo et al, 2014; Di et al, 2015)."*

For further clarification, the NCEP data (GFS) used in this study is the forecast data which is initialised at 00hUTC on the day. The GTS data are not assimilated in NCEP GFS, which has been proven by the improvements of the rainfall forecasts from the data assimilation modes with the GTS data assimilated in the outer domain.

**Point 3:** Data assimilation is routinely done at various levels in numerical weather prediction. The big problem to produce a hydrologically compatible rainfall forecast is that many of those forecasts fail to capture the two essential aspects: amount and distribution. With reference to the paper, Fig. 5 shows a

consistent time shift of all the runs in all modes, i.e., the predicted storms started and stopped around 6-h earlier than the actual one. This might be linked to the setting of assimilation, and I suspect that more likely than not it is due to the constraint imposed by the background field from the lateral boundary conditions. This however, has not been properly explored.

**Reply:** We thank the referee's close observation and agree with his opinion. The time shift of the storm was more likely caused by the background field from the lateral boundary conditions. From Fig. 5 it can be seen that the 6h shift starts with run1, which is the original WRF run without data assimilation. But there are also some studies indicating that when data containing the information of water vapor are assimilated in the numerical weather prediction model, the predicted rainfall may start and stop earlier than the actual one (Georgakakos, 2000; Sun et al, 2016). The main reason is likely that the information of water vapor can make the rain in the initial fields form and fall to the earth more quickly (Sun, 2005). In this study, both the radar reflectivity and the GTS data contain the information of water vapor, and the radial velocity did no help in improving the dynamic field by data assimilation. All the above reasons resulted in a consistent 6h time shift of all the runs in Fig. 5. To deal with this, an error prediction model could be built to correct the consistent error. Some studies also suggest the assimilation of the latent heat might help improve the start and ending time of the forecasted rainfall process (Stephan et al, 2010; Schraff et al, 2016). The time shift issue is now addressed in the manuscript by adding the following paragraph in the Line 20-25, Page 10:

*"It can be found in Fig. 5 that the predicted storms always start and end around 6-h earlier than the observations. Besides the errors in the boundary conditions, it is found that the assimilation of the water vapor information (contained in the radar reflectivity and the GTS data) can make the rain in the initial fields form and fall to the earth more quickly (Georgakakos, 2000; Sun, 2005; Sun et al, 2016). Considering the error is consistent, an error prediction model could be built in further studies, and the assimilation of the latent heat may also be helpful in correcting the starting and ending time of the forecasted rainfall process (Stephan et al, 2010; Schraff et al, 2016)."*

**Point 4:** The choice of using cumulative (only) rainfall may be OK to compare the overall amount in general. Again, for hydrological use, we'd like to see how the prediction agrees with the distributions (both temporal and spatial) of the actual rainfall. So, I think it would be interesting to have a normal hyetograph and a spatial distribution would be more helpful. Some derivative indices like RMSE would make the discussion more convincing.

**Reply:** The referee's suggestion is thoughtful. Both the temporal and spatial distributions of the rainfall can have potential impacts on the formation of the flow, thus are paid equally important attention by the hydrologists. The spatial distributions as well as the temporal variations of the forecasted rainfall from different data assimilation modes are added in the new Section 4.3. Two figures (Fig. 6 and Fig. 7) are added to respectively present the hyetographs and the spatial distributions. The index RMSE is used to evaluate the spatial and temporal distributions of the forecasted rainfall against the rain gauge observations, as shown in Table 4.

[revised manuscript text omitted]

**Point 5:** A few terminology and grammar issues: 1) we don't quite often use 'curve' in general, hyetograph is a better and more accurate choice when being used to describe the temporal distribution of rainfall. 2) P5 L26-28 'If more than … average value'. This sentence is confusing. 3) P11 L13-15 'The assimilation of radar velocity …' I think you meant 'radar radial velocity'. Also the sentence itself is self-contradicting: moisture transport does affect the rainfall 'physical' process. Please elaborate more. 4) P11 L18 '… are quite variably' should be 'are quite variable'.

**Reply:** The terminology and grammar errors are carefully checked and corrected in the revised manuscript. Efforts are also made to further improve the readability of the paper. The four issues mentioned above are addressed as follows:

1) The terminology 'curve' is replaced by 'hyetograph' throughout the manuscript.

2) The sentence is revised as: *"The forecasted areal rainfall was calculated by averaging values of the grid cells those have more than 50% area located inside the Zijingguan catchment."*

3) The sentence is revised as: *"The assimilation of radar radial velocity cannot directly influence the physical process of rainfall formation, although the assimilation can change the wind field and affect the water vapor transport."*

4) Revised accordingly.

---

## Author Response (AR2)

**Evaluation of Doppler radar and GTS data Assimilation for NWP Rainfall Prediction of an Extreme Summer Storm in Northern China: from the Hydrological Perspective**

**(Manuscript ID: hess-2017-689)**

**Technical Corrections**

**Point 1:** P4 L23-24: '…always performed well' is this really the case or should it be limited to the area you studied?

**Reply:** The sentence is revised as follows in P4 L23-24 and a reference is also added:

*'…the horizontal grid spacing of the WRF inner domain (Domain 2) was set to 3 km, and the downscaling ratio was set to 1:3, which was commonly used and always performed well in the Beijing-Tianjin-Hebei region of northern China (Liu et al, 2012; Chambon et al, 2014; Tian et al, 2017b).'*

Reference:

Tian, J., Liu, J., Yan, D., Li, C., Chu, Z., and Yu, F.: An assimilation test of Doppler radar reflectivity and radial velocity from different height layers in improving the WRF rainfall forecasts, Atmos. Res., 198, 132-144, doi: 10.1016/j.atmosres.2017.08.004, 2017b.

**Point 2:** P4 L24-25 'The large scale topography and … can be covered by the nested domain…' I think it's better to say your settings have been shown to be effective in previous studies in terms of representing …

**Reply:** The sentence is revised accordingly in P4 L24-26:

*'The settings of the nested domains have been shown to be effective in previous studies in terms of representing the large scale topography and the main climate characteristics in the study area (Wang et al, 2013; Tian et al, 2017b).'*

**Point 3:** P5 L17 'The total water mixing ratio …' You seem to try to highlight your choice of using qt but don't give a proper reason for doing this. Is such choice better/more effective or just a random choice?

**Reply:** The use of the total water mixing ratio is not a random choice, but a fixed setting in WRF-3DVar for the assimilation of the radar reflectivity data. The total water mixing ratio has a more direct relation with the radar reflectivity compared to the pseudo-relative humidity, and has been proven to be effective in WRF-3DVar (Sun and Crook, 1997).

The reason for using the total water mixing ratio as the control variable is restated in P5 L19-23 to

avoid misunderstanding:

*'In the WRF-3DVar system, the total water mixing ratio $q_t$ was used as the moisture control variable instead of the pseudo-relative humidity when assimilating the radar reflectivity data (Dudhia, 1989). The water mixing ratio has a more direct relation with the radar reflectivity, as described by Eq. (2), which has been proven to be effective in WRF-3DVar as an observation operator to calculate the model-derived radar reflectivity Z from the rainwater mixing ratio $q_r$ (Sun and Crook, 1997).'*

$$Z = 43.1 + 17.5\log(\rho q_r)$$
(2)

**Point 4:** P24 Table 1 Column 'Number of the data', please use a more proper/accurate expression. The word 'data' itself is plural and we don't say one data or two data.

**Reply:** The expression is replaced with 'number of observations'.